



# Comparison of occurrence-bias-adjusting methods for hydrological impact modelling

Jorn Van de Velde[1,2], Bernard De Baets[2], Matthias Demuzere[3,1], and Niko E. C. Verhoest[1]

[1]Hydro-Climatic Extremes Lab, Ghent University, Ghent, Belgium
[2]KERMIT, Department of Data Analysis and Mathematical Modelling, Ghent University, Ghent, Belgium
[3]Department of Geography, Ruhr-University Bochum, Bochum, Germany

**Correspondence:** Jorn Van de Velde (jorn.vandevelde@ugent.be)

**Abstract.** Over the past decade, various methods for bias adjustment of precipitation occurrence or intensity have been proposed. However, the performance of combined methods has not yet been thoroughly evaluated, especially in a hydrological and climate change context. In this study, four occurrence-bias-adjusting methods are combined with one univariate and one multivariate intensity-bias-adjusting method. The occurrence-bias-adjusting methods include thresholding, Stochastic Singularity

Removal, Triangular Distribution Adjustment, and are compared with the intensity-bias-adjusting methods without specific adjustment as a baseline. These combined methods are compared with respect to precipitation amount, precipitation occurrence and discharge. This comparison, summarized in terms of the residual bias relative to both the observations and the model bias, shows significant differences in performance. Occurrence-bias-adjusting methods that add stochasticity perform worse, an effect that is reinforced by multivariate intensity-bias-adjusting methods. The use of simpler methods is thus advised until

the uncertainty caused by combining methods is better understood.

## 1 Introduction

Climate change is one of the largest threats currently faced by society, with impacts on many (eco)systems. These impacts are caused by the increase in naturally occurring hazards, such as droughts, wildfires, hurricanes and floods (IPCC, 2012).

To assess how these hazards are influenced by climate change conditions, a modeling chain consisting of Global Circulation Models (GCMs), Regional Climate Models (RCMs) and local impact models is commonly used (Maraun et al., 2010). The GCMs are applied to generate global future time series (Eyring et al., 2016). However, their scale is too coarse to be used directly in local impact models. This is especially true in hydrology, where local meteorological variables such as precipitation can differ substantially within a watershed. Therefore, RCMs are used to downscale the global data to a local scale. This

is conducted by physically simulating the local climate, using the GCM's output as boundary conditions. RCMs with a grid resolution of 10 km are commonly used, enhancing the amount of available information (Jacob et al., 2014). Nevertheless, some systematic errors still occur in these models, due to imperfect parametrization, discretization and spatial averaging (Teutschbein





and Seibert, 2012). These errors lead to the presence of biases, most often in precipitation (Kotlarski et al., 2014). Recently, convection-permitting models, which have a grid resolution of up to 1 km have become available (Prein et al., 2015; Kendon

et al., 2017; Helsen et al., 2019). However, these models have a high computational demand and their implementation is therefore no common practice yet in impact studies. The biases in coarser RCMs are generally described as *"a systematic difference between a simulated climate statistic and the corresponding real-world climate statistic"* (Maraun, 2016), though there is some debate on the use of the term 'bias' (Ehret et al., 2012). Biases increase the uncertainty in the last step of the modeling chain, i.e. the local impact modelling, thus entailing the necessity to adjust them (Fowler et al., 2007; Christensen

et al., 2008; Maraun, 2016). Until recently, the term 'bias correction' was prominently used, but to emphasize that most of these methods cannot completely remove the bias, the term 'bias adjustment' has gained popularity. In this paper, we follow this recent trend (Vrac et al., 2016; Maraun et al., 2017; Vrac, 2018; Räty et al., 2018; Zscheischler et al., 2019) and thus refer to 'bias adjustment'.

During the last 15 years, many methods have been developed to overcome the above-mentioned bias problem (see Teutschbein

and Seibert (2012); Gutiérrez et al. (2019) for recent overviews). These methods, which adjust the mean, variance and/or the full distribution of the variable under consideration, are well-studied for hydrological impact studies (e.g. Addor and Seibert (2014); Räty et al. (2018)) and use a 'transfer function' to transfer the information from the observations to the simulations. However, an overlooked aspect in many of these methods is the adjustment of rainfall occurrence. Climate models often simulate too many rainy days, the so-called drizzle effect (Gutowski et al., 2003; Argüeso et al., 2013) and sometimes too many

dry days (Themeßl et al., 2012). This has since long been acknowledged to be a problem, especially when the intermittence of rainfall is important (Ines and Hansen, 2006), as is the case for hydrological impact assessment. A rainy day may cause an increase in soil moisture. As a consequence, the infiltration capacity decreases, leading to more runoff and riverflow, and possibly leading to flooding. Though most bias-adjusting methods implement some basic form of frequency correction before adjusting the bias in intensity, there is limited research on the effect of these methods. One of the earliest studies on the per-

formance of bias adjustment with respect to transition probabilities was published by Rajczak et al. (2016), while Vrac et al. (2016) published one of the first studies on the comparison of occurrence-bias-adjusting methods. The latter proposed a new occurrence-bias-adjusting method, named Stochastic Singularity Removal (SSR). This method was shown to perform well, but has only been compared with few other methods. As various other occurrence-bias-adjusting methods are also available and seem to perform well, the goal of this paper is to compare a selection of these methods and study how they interact with

the subsequent intensity-bias-adjusting methods. Most of the common bias-adjusting methods, and therefore also the methods used in this comparison, stem from the quantile-mapping-method family (Panofsky et al., 1958). Quantile mapping was found to be the best-performing method, especially with respect to the simulation of hydrological time series (Rojas et al., 2011; Gudmundsson et al., 2012). This method adjusts the full distribution of the variable's future simulations on the basis of cumulative distribution functions (CDFs) of the historical simulations (hs) and the historical observations (ho). A transfer function

is composed from these CDFs, resulting in an adjusted future simulation (fa) for a day $i$ in the time series:

$$x_i^{\mathrm{fa}} = F_{x^{\mathrm{ho}}}^{-1}\left(F_{x^{\mathrm{hs}}}\left(x_i^{\mathrm{fs}}\right)\right),\tag{1}$$





with fs the index of the uncorrected future simulations, $F$ a CDF, and $F_x$ the corresponding CDF value for a certain quantile value $x$. This process is shown in Fig. 1.

**Figure 1.** Conceptualization of the quantile mapping method. Every $x^{\text{fs}}$ (star) is replaced by a $x^{\text{fa}}$ (circle) following Eq. (1), nudging the future simulations closer to the supposed future observations.

Various extensions and variants of the standard quantile mapping have been proposed. A recent variant is Quantile Delta
Mapping (QDM) (Li et al., 2010; Wang and Chen, 2014; Cannon et al., 2015). It was introduced to better retain the trends of the climate model and though being critized (Switanek et al., 2017), it is increasingly used (Mehrotra and Sharma, 2016; Nguyen et al., 2016; Cannon, 2018). Older quantile mapping methods are also still in use, such as CDF-t (Michelangeli et al. (2009), used in e.g. Vrac (2018)), and standard empirical quantile mapping, used in e.g. Räty et al. (2018), Zscheischler et al. (2019). Nevertheless, given its generally good performance, QDM was chosen as the intensity-bias-adjusting method in this
comparison study.



During the last few years, multivariate intensity-bias-adjusting methods have received considerable attention. The development of these methods became important after some authors began to question the inability of univariate methods such as QDM to adjust biases in the inter-variable correlation of climate models (Hagemann et al., 2011; Wilcke et al., 2013; Hewitson et al., 2014). Subsequently, many multivariate methods have been proposed (e.g. Li et al. (2014); Mao et al. (2015); Vrac and Friederichs (2015); Mehrotra and Sharma (2016); Dekens et al. (2017); Cannon (2018); Nguyen et al. (2018)). As these methods have increased the complexity of bias adjustment, there is a growing need to study the combined effect of an occurrence-bias-adjusting method and a subsequent multivariate intensity-bias-adjusting method. To the authors' knowledge, the combination of occurrence-bias-adjusting and multivariate intensity-bias-adjusting methods has not yet been investigated. To make an initial assessment, one multivariate method is chosen to be combined with the different occurrence-bias-adjusting techniques, namely the Multivariate Bias Correction in $n$ dimensions (MBCn) method (Cannon, 2018). This method is chosen as it is one of the more commonly implemented multivariate bias-adjusting methods (Räty et al., 2018; Meyer et al., 2019; Zscheischler et al., 2019). Additionally, it only differs from the QDM method in the final step, where a rank-based shuffling is applied. All differences in performance may thus be linked to this step.

The use of both a univariate and a multivariate intensity-bias-adjusting method will facilitate a more complete comparison of the occurrence-bias-adjusting methods and their interaction with other methods. The methods compared in this paper include the combination of three occurrence-bias-adjusting methods with respectively a classic univariate intensity-bias-adjusting method, i.e. QDM, and a multivariate intensity-bias-adjusting method, i.e. MBCn. The three occurrence-bias-adjusting methods include an extended thresholding method, a stochastic thresholding method using the triangular distribution, and the SSR method. These can all be considered more advanced methods. Simpler methods, such as the 'positive correction' (Mao et al., 2015) and the 'threshold adaptation' method (Schmidli et al., 2006) were proven to be outperformed by SSR (Vrac et al., 2016). To enable a baseline comparison, the intensity-bias-adjusting methods are also applied without any specific occurrence-bias-adjustment, which corresponds to the direct approach as also implemented in Vrac et al. (2016). The intensity-bias-adjusting methods used here inherently apply a basic occurrence-bias-adjustment: if the observations have more dry days than the simulations, the days with the lowest precipitation amounts are automatically mapped to zero precipitation due to the quantile mapping. It also works the other way around: if the observations have less dry days, then days with a low precipitation depth are mapped to a higher precipitation amount.

The comparison in this paper is performed by calibrating on historical time series and validating on recent past time series. In order to perform a robust calibration and validation, the time series used has to be long enough (Berg et al., 2012). In Belgium, the Royal Meteorological Institute collects precipitation measurements in Uccle since the late 19th century. This time series has been used in multiple studies (Verhoest et al., 1997; De Jongh et al., 2006; Verstraeten et al., 2006; Vandenberghe et al., 2011; Willems, 2013) and will also be used in this study. All the combined methods will be compared using this rainfall time series as observations (Demarée, 2003) to adjust EURO-CORDEX RCM simulations (Jacob et al., 2014). The time series is used for the period 1970-1989 for calibration and for the period 1998-2017 for validation. This allows for the assessment of the effect of climate change trends on bias adjustment, as there is a visible impact in the validation time window (IPCC, 2013). If the data are used in this way, it is possible to gain understanding of how the bias-adjusting methods will perform under



real future conditions, e.g. from 2070-2100, a period that is often used to study future conditions. To evaluate the precipitation adjustment thoroughly, indices have been chosen for the precipitation amount, its occurrence and the resulting discharge. The use of these indices permits a clear comparison of the combined methods and their performance.

To summarize, the goal of this study is to compare the performance and hydrological impact of eight different combinations of occurrence- and intensity-bias-adjusting methods in a climate change context. This paper is structured as follows. In Section 2, the observational data and climate model simulations are addressed. In Section 3, the occurrence- and intensity-bias-adjusting methods are explained together with the evaluation and calculation set-up. In Section 4, a novel visualisation method, based on residual biases is explained and used to present the results, with a discussion in Section 5 and a conclusion in Section 6.

## 2 Data

### 2.1 Observations

The observational data used were obtained from the Belgian Royal Meteorological Institute (RMI) Uccle observatory. The most important time series used is the 10-min precipitation amount, gauged with a Hellmann-Fuess pluviograph, from 1898 to 2018. An earlier version of this precipitation dataset was described by Demarée (2003) and analyzed in De Jongh et al. (2006). The 10-min precipitation time series was aggregated to daily level to be comparable with the other time series used. As the last complete year was 2017, the data were used from 1901 to 2017, amounting to 117 years of daily data.

For the multivariate method, the precipitation time series was combined with a 2 meter air temperature and potential evaporation time series. The daily potential evaporation was calculated by the RMI from 1901 to 2019, using the Penman formula for a grass reference surface (Penman, 1948) with variables measured at the Uccle observatory. Daily average temperatures were obtained using measurements from 1901 to 2019. Limited by the length of the precipitation time series, the same 117 years of data as for the univariate method were used.

### 2.2 Climate simulations

For the simulations, data from the EURO-CORDEX project (Jacob et al., 2014) were used. The Rosby Centre regional climate model RCA4 was used (Strandberg et al., 2015) as it is one of the few RCMs with potential evaporation as an output variable. This RCM was forced with boundary conditions from the MPI-ESM-LR GCM (Popke et al., 2013). Historical data and scenario data for the grid cell comprising Uccle were respectively obtained for 1970-2005 and 2006-2100. The former time frame is limited by the earliest available data from the RCM. The latter time frame was only used until 2017, in accordance with the observational data. As climate change scenario, an RCP4.5 forcing was used in this paper (Van Vuuren et al., 2011). Since only 'near future' (from the model point of view) data were used, the choice of forcing does not have a large impact. However, when studying scenarios in a time frame further away from the present, using an ensemble of forcings is more relevant to be aware of the uncertainty regarding future climate change impact. Evaluations of the RCA4 model have shown that there is a bias in





precipitation, especially in winter (Strandberg et al., 2015), but this bias is in line with the biases from other EURO-CORDEX models (Kotlarski et al., 2014).

As Uccle (near Brussels) is situated in a region with small topographic differences, it is assumed that the conditions in Uccle
can be applied anywhere within the climate model grid cell and the variance within this cell is about the same. This assumption can be made as long as the resulting adjusted data are not used for extremely localized studies, such as urban hydrology impact assessment.

## 3   Bias-adjusting methods

### 3.1   Thresholding

Thresholding is one of the most common occurrence-bias-adjusting methods and has been in use for many years (e.g. Hay and Clark (2003); Schmidli et al. (2006); Ines and Hansen (2006)). This method assumes more wet days in the simulations than in the observations (Vrac et al., 2016; Themeßl et al., 2012), an idea based on the drizzle effect of RCMs (Gutowski et al., 2003). For Belgium, this assumption holds: the drizzle effect generally occurs in RCMs in north-western Europe (Themeßl et al., 2012). The simplest form of thresholding transforms all values in the climate model simulations below a certain threshold
to 0. A more advanced method, used e.g. in Switanek et al. (2017), ensures that the number of days below the threshold in the observed and simulated time series is the same.

To adjust the number of wet and dry days, the frequency of dry days in the observations and in the simulations is calculated. The difference between the two frequencies, $\Delta N$, is the number of days of the simulation time series that has to be adapted. This adaption is done by sorting the wet days in increasing order of precipitation amount and thus only changing the $\Delta N$
lowest days of the simulation time series by setting them to 0.

In this advanced version of thresholding, some considerations are made. First, a day is considered wet if its simulated precipitation amount is above 0.1 mm, to account for measurement errors in the observations. Second, the adjustment is done on a monthly basis, to withhold a realistic temporal structure. This implies that to correct the number of wet days in month $m$, all days of month $m$ of the time series are selected. Third, both historical and future simulations are adjusted at the same
moment, to ensure a sound comparison during the intensity phase of the adjustment. If only either the historical or future time series would have been adjusted, the assumption that the bias can be transferred from the historical to the future time period would be impaired. The thresholding method is summarized in Algorithm 1.





---

**Algorithm 1** Thresholding

**Input:**

Historical observations $X^{\mathrm{ho}}$

Historical simulations $X^{\mathrm{hs}}$

Future simulations $X^{\mathrm{fs}}$

**Output:**

Adjusted historical $X^{\mathrm{hs}}_{\mathrm{out}}$ and future simulations $X^{\mathrm{fs}}_{\mathrm{out}}$

 

Initialization

**for** $m = 1 : 12$ **do**

    Select data for month $m$: $X^{\mathrm{ho}}_m$, $X^{\mathrm{hs}}_m$ and $X^{\mathrm{fs}}_m$ {Loop over months}

    Calculate the percentage of dry days in month $m$ for the historical observations

    Calculate the percentage of dry days in month $m$ for the historical simulations

    Calculate $\Delta N$ for month $m$

    {Adjustment of historical simulated time series}

    Select and sort the wet days

    Set the $\Delta N$ wet days with the lowest precipitation amount to 0

    Restore the original order of the wet days of month $m$

    Restore the full historical time series for month $m$

    {Adjustment of future simulated time series}

    Select and sort the wet days

    Set the $\Delta N$ wet days with the lowest precipitation amount to 0

    Restore the original order of the wet days of month $m$

    Restore the full future time series for month $m$

    {Reconstruction}

    Replace the data in $X^{\mathrm{hs}}_{\mathrm{out}}$ with the adjusted data for month $m$

    Replace the data in $X^{\mathrm{fs}}_{\mathrm{out}}$ with the adjusted data for month $m$

**end for**

---

## 3.2 Stochastic Singularity Removal

To overcome the assumption that the simulated time series has to have more wet days than the observations, Vrac et al. (2016)

proposed the Stochastic Singularity Removal (SSR) method. This method temporarily removes the zeroes from all time series and reintroduces the zeroes after the bias adjustment is done, based on the frequency method mentioned in Cannon et al. (2015) and a method used to alter temperature data in Zhang et al. (2009).





The removal of zeroes is done by calculating the lowest amount of rain $P_{min}$ in any of the time series used, including the historical observations, historical simulations and future simulations. If days with a precipitation amount below 0.1 mm are

considered to be dry days (based on the measurement error, see Section 3.1), $P_{\min}$ is the lowest amount of precipitation above 0.1 mm. As Vrac et al. (2016) mention, the use of a precipitation threshold for dry days does not have any implications, but is implemented here to be in line with other implementations of the measurement error. For all days with rainfall amounts below $P_{\min}$, a new rainfall depth $P_{\text{new}}$ is randomly drawn from a uniform distribution on $[0.1, P_{\min}]$. The random simulation also adds a stochastic element to this form of bias adjustment, which some authors have noted to be necessary (Maraun, 2013; Mao

et al., 2015; Vrac, 2018). By using this method, the subsequent intensity-bias adjustment will be applied on all days, as if they were all wet days. There are thus no longer dry days, or singularities, in the time series, hence the method's name. After the application of the intensity-bias adjustment, a last post-processing step is applied: all days of $X^{\text{fs}}$ with a rainfall amount below $P_{\min}$ are set to zero. The full procedure is summarized in Algorithm 2.

---

**Algorithm 2** SSR

**Input:**

    Historical observations $X^{\text{ho}}$

    Historical simulations $X^{\text{hs}}$

    Future simulations $X^{\text{fs}}$

**Output:**

    Adjusted historical $X^{\text{hs}}_{\text{out}}$ and future simulations $X^{\text{fs}}_{\text{out}}$

    {Before the intensity-bias adjustment}

    Determine the length of the time series Ndays

    Determine $P_{\min}$ based on $X^{\text{ho}}$, $X^{\text{hs}}$ and $X^{\text{fs}}$

    **for** $i = 1 : \text{Ndays}$ **do**

        **if** $X^{\text{ho}}(i) < P_{\min}$ **then**

            Simulate a new value

        **end if**

        **if** $X^{\text{hs}}(i) < P_{\min}$ **then**

            Simulate a new value

        **end if**

        **if** $X^{\text{fs}}(i) < P_{\min}$ **then**

            Simulate a new value

        **end if**

    **end for**

    {After the intensity-bias adjustment}

    Set all $X^{\text{fs}} < P_{\min}$ to 0

---





### 3.3 Triangular Distribution Adjustment

The Triangular Distribution Adjustment method (TDA), as developed by Pham (2016), is a more complex method. In the first step, the number of days to be removed from or added to the future time series is calculated using the ratio of the historical observation and simulation dry day frequency respectively $f^{\mathrm{ho}}$ and $f^{\mathrm{hs}}$. This ratio is assumed to be the same for the future scenarios such that it can be used to calculate a corrected future dry day frequency, $f^{\mathrm{fa}}$, which is in turn used to calculate the number of days $\Delta N$ to be removed or added.

To make sure that the removal or addition of dry days does not change the extremes and adds stochasticity (noted to be necessary by Maraun (2013); Mao et al. (2015); Vrac (2018)), the triangular distribution, hence the method's name, is used. The general idea of this method is given in Figure 2. The CDF of the triangular distribution is given by

$$T(x) = \begin{cases} 1 - \frac{(b-x)^2}{b^2} & \text{if } 0 \leq x < b \\ 1 & \text{if } b \leq x \end{cases} \tag{2}$$

The single parameter of this triangular distribution, $b$, is based on a threshold value $x_{\mathrm{thr}}$, to ensure that extreme values are not

removed, and is calculated as follows:

$$b = F_{x^{\mathrm{fs}}}(x_{\mathrm{thr}} \mid x > 0.1), \tag{3}$$

with $F_{x^{\mathrm{fs}}}(x \mid x > 0.1)$ the CDF of the precipitation of wet days.

   Dry days are added as follows:

1. For every wet day to be removed, chose a day $t$ (with precipitation $x_t > 0.1$) randomly from the wet day time series.
This day has a corresponding cumulative probability of $\xi = F_{x^{\mathrm{fs}}}(x_t \mid x > 0.1)$.

2. Sample $k$ from a uniform distribution on $[0,1]$. If $T(\xi) < k$, then draw $x_t$ from a uniform distribution on $[0,0.1]$ to replace a rainy day with a dry day. $x$ is drawn randomly, to take into account that model simulations hardly have zero values.

3. If $T(\xi) > k$, then repeat from step 1.

If dry days need to be removed, this is done using the value of $\xi$, which is calculated using Eq. (2):

$$T(\xi) = 1 - \frac{(b-\xi)^2}{b^2} \text{ from which } \xi = b\left(1 - \sqrt{1 - T(\xi)}\right). \tag{4}$$

   In this case, there is no restriction on the choice of days, except that they have to be dry. Thus $\Delta N$ dry days will be randomly selected from the time series and removed, without considering the temporal structure of the time series. The process of dry day removal is given as follows:



1. For every dry day to be removed, choose a dry day $t$ randomly from the dry day time series (with $x_t < 0.1$) and sample $k$ from a uniform distribution on $[0, 1]$.

2. $T(\xi) = k$ and thus $\xi = b\left(1 - \sqrt{1-k}\right)$.

3. Then set $x_t = F_{x^{\mathrm{fs}}}^{-1}(\xi) = F_{x^{\mathrm{fs}}}^{-1}\left(b\left(1 - \sqrt{1-k}\right) \mid x > 0.1\right)$.

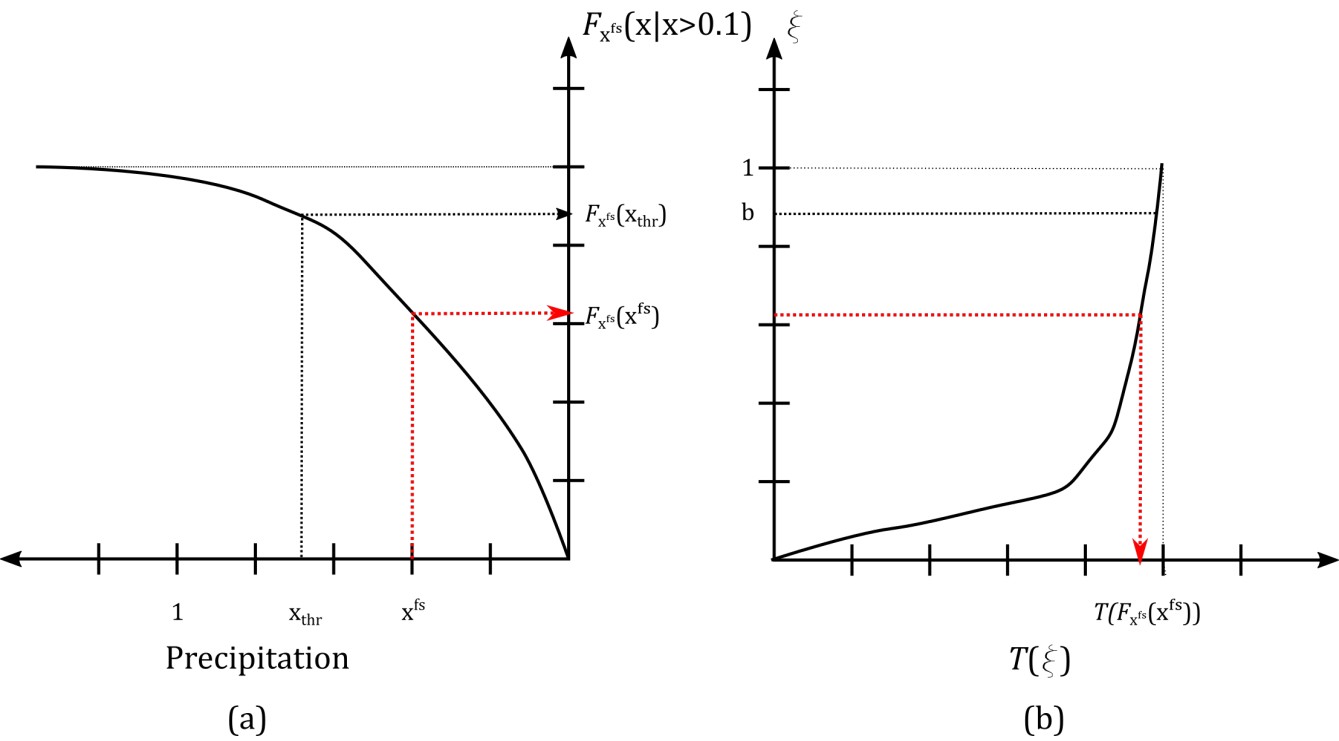

**Figure 2.** Overview of the distributions used in TDA: (a) CDF of the precipitation of the wet days in the future simulation, (b) CDF of the triangular distribution. Adapted from Pham (2016).

  As in thresholding, this method is applied on a monthly basis and for both historical and future simulations, which results in
Algorithm 3.





---

**Algorithm 3** Triangular Distribution Adjustment

---

**Input:**

Historical observations $X^{\mathrm{ho}}$

Historical simulations $X^{\mathrm{hs}}$

Future simulations $X^{\mathrm{fs}}$

**Output:**

Adjusted future simulations $X^{\mathrm{fs}}_{\mathrm{out}}$

Initialization

**for** $m = 1:12$ **do**

    Select data for month $m$: $X^{\mathrm{ho}}_m$, $X^{\mathrm{hs}}_m$ and $X^{\mathrm{fs}}_m$ {Loop over months}

    Calculate $f^{\mathrm{hs}}$, $f^{\mathrm{ho}}$ and $f^{\mathrm{fs}}$ {Begins dry day frequency calculation}

    Calculate $f^{\mathrm{fa}} = f^{\mathrm{fs}} \cdot f^{\mathrm{ho}}/f^{\mathrm{hs}}$

    Calculate the difference in dry days $\Delta N$

    Select the wet days for month $m$ {Empirical CDF}

    Select the dry days for month $m$

    Calculate the empirical CDF $F_{x^{\mathrm{fs}}}$ for the wet days

    {Adjustment}

    **if** $\Delta N > 0$ **then**

        {Begins addition of dry days} **while** counter $\leq \Delta N$ **do**

        Randomly select a day $t$

        Calculate $\xi$ using $x_t$

        Select a random value k

        **if** $T(\xi) < k$ **then**

            Randomly replace the value of day $t$ with a value on [0,0.1]

            Add 1 to the counter

        **end if**

      **end while**

    **else if** $\Delta N < 0$ **then**

        {Begins Removal of dry days}  Set counter to 1 **while** counter $\leq |\Delta N|$ **do**

        Randomly select a day $t$

        Calculate $\xi$

        Calculate $x_t$

        Add 1 to the counter

      **end while**

    **end if**

    Recombine the wet and dry day time series in the original order for the month $m$

    Reintroduce the adjusted data for month $m$ in the original time series

**end for**





Note that Algorithm 3 is easily adapted for historical simulations by calculating $\Delta N$ using only $f^{\text{hs}}$ and $f^{\text{ho}}$, in the same way as in Algorithm 1. The parameter $b$ was set to 0.9, as this ensures that the highest extremes are never changed to dry days, while not restricting the choice of days.

### 3.4 Quantile Delta Mapping

The Quantile Delta Mapping (QDM) method was first proposed by Li et al. (2010), with the idea of preserving the climate simulation trends and taking some non-stationarity into account, and was extended by Wang and Chen (2014) to be able to handle precipitation adjustment better. A comparison with other quantile mapping methods by Cannon et al. (2015) proved this method to outperform others with respect to the preservation of trends. Cannon et al. (2015) bundled both the methods by Li et al. (2010) (Equidistant CDF-matching) and Wang and Chen (2014) (Equiratio CDF-matching) under the name 'Quantile

Delta Mapping'. This name was chosen because of the link with other delta methods, as presented in Olsson et al. (2009), Willems and Vrac (2011) and Räty et al. (2014). Consequently, the name QDM will also be used in this paper.

Mathematically, this method can be written as

$$x_i^{\text{fa}} = x_i^{\text{fs}} + F_{x^{\text{ho}}}^{-1}\left(F_{x^{\text{fs}}}\left(x^{\text{fs}}\right)\right) - F_{x^{\text{hs}}}^{-1}\left(F_{x^{\text{fs}}}\left(x^{\text{fs}}\right)\right) \tag{5}$$

in the additive case and

$$x_i^{\text{fa}} = x_i^{\text{fs}} \frac{F_{x^{\text{ho}}}^{-1}\left(F_{x^{\text{fs}}}\left(x^{\text{fs}}\right)\right)}{F_{x^{\text{hs}}}^{-1}\left(F_{x^{\text{fs}}}\left(x^{\text{fs}}\right)\right)} \tag{6}$$

in the ratio or multiplicative case. In this paper, the additive version is used for temperature time series and the multiplicative for precipitation time series. This choice is based on the work of Wang and Chen (2014), who have shown that using the additive version for precipitation results in unrealistic precipitation values and introduced a multiplicative version.

To ensure the consistency of the time series, Themeßl et al. (2011) implemented a 61-day moving window. Here, a 91-day

moving window is opted for, as suggested by Rajczak et al. (2016) and Reiter et al. (2018). This enables the adjustment of each day based on $91\,\text{days/year} \cdot 20\,\text{years} = 1820$ days. These days were used to build an empirical CDF (as in Gudmundsson et al. (2012); Gutjahr and Heinemann (2013), among others), because of the ease of application. It is also important to note that for precipitation and evaporation, the adjustment was calculated and applied only on the days considered wet, e.g. with a precipitation higher than 0.1 mm.

### 3.5 Multivariate Bias Correction in $n$ dimensions

About 10 years ago, it became clear that multivariate correlation adjustment was necessary, as different studies started pointing out that univariate methods could never directly adjust the correlation bias in climate models (Hagemann et al., 2011; Wilcke et al., 2013; Hewitson et al., 2014). This problem encouraged the publication of many studies describing multivariate bias-adjusting methods (e.g. Li et al. (2014); Mao et al. (2015); Vrac and Friederichs (2015); Mehrotra and Sharma (2016); Dekens



et al. (2017); Cannon (2018); Nguyen et al. (2018)). These methods all use different approaches to describe the dependence between variables. However, Cannon (2018) recently pointed out that many of these approaches are either too limited to adjust dependence, as in Mehrotra and Sharma (2016) or Cannon (2016), or that they make strong stationarity assumptions, as in Vrac and Friederichs (2015). Consequently, Cannon (2018) proposed a novel method, based on the N-dimensional probability density function transform (N-pdft), from the field of image processing and computer vision (Pitié et al., 2005, 2007). Instead of adjusting pixels with an RGB colour combination, this method is now used for adjusting time steps with meteorological variables. Combining this with the trend-preserving QDM, Cannon named this multivariate bias-adjusting method 'Multivariate Bias Correction in $n$ dimension' (MBCn). Though this paper follows the recent trend to use 'adjust' instead of 'correct', the name 'MBCn' will be used for clarity and clear reference to the original method.

In brief, the method consists of three steps that are applied iteratively until convergence is reached and a fourth step to restore the trends. These steps are applied on the different types of data: historical observations of temperature, evaporation and precipitation (combined in the matrix $\mathbf{X}^{\mathrm{ho}}$), historical climate model simulations of the three variables (the matrix $\mathbf{X}^{\mathrm{hs}}$) and climate model projections of the three variables (the matrix $\mathbf{X}^{\mathrm{fs}}$), which have to be adjusted. All these datasets are of size $T \times N$, with $T$ the number of time steps and $N$ the number of variables. As there were no prior examples of evaporation bias-adjustment, it was assumed that this variable could be adjusted in the same manner as precipitation, i.e. a multiplicative adjustment was used and only 'wet days' (evaporation > 0.1 mm) were adjusted.

Considering the $j$-th iteration of the method, denoted by the subscript $[j]$, the first step consists of rotating the data sets using an $N \times N$ randomly generated orthogonal rotation matrix $\mathbf{R}_{[j]}$. This orthogonal rotation matrix was created using the algorithm by Mezzadri (2007, pg. 597). This rotation is formulated as

$$
\begin{aligned}
\tilde{\mathbf{X}}_{[j]}^{\mathrm{hs}} &= \mathbf{X}_{[j]}^{\mathrm{hs}} \mathbf{R}_{[j]} \\
\tilde{\mathbf{X}}_{[j]}^{\mathrm{fs}} &= \mathbf{X}_{[j]}^{\mathrm{fs}} \mathbf{R}_{[j]} \\
\tilde{\mathbf{X}}_{[j]}^{\mathrm{ho}} &= \mathbf{X}_{[j]}^{\mathrm{ho}} \mathbf{R}_{[j]}
\end{aligned}
\tag{7}
$$

with $\tilde{\mathbf{X}}_{[j]}$ the resulting rotated matrix. In the next step, quantile delta mapping is applied to each variable in $\tilde{\mathbf{X}}_{[j]}^{\mathrm{hs}}$ and $\tilde{\mathbf{X}}_{[j]}^{\mathrm{fs}}$, using the corresponding variable in $\tilde{\mathbf{X}}_{[j]}^{\mathrm{ho}}$ as the target. The resulting matrices $\mathbf{X}_{[j]}^{\mathrm{hc}}$ and $\mathbf{X}_{[j]}^{\mathrm{fa}}$ are rotated back:

$$
\begin{aligned}
\mathbf{X}_{[j+1]}^{\mathrm{hs}} &= \mathbf{X}_{[j]}^{\mathrm{hc}} \mathbf{R}_{[j]}^{-1} \\
\mathbf{X}_{[j+1]}^{\mathrm{fs}} &= \mathbf{X}_{[j]}^{\mathrm{fa}} \mathbf{R}_{[j]}^{-1} \\
\mathbf{X}_{[j+1]}^{\mathrm{ho}} &= \mathbf{X}_{[j]}^{\mathrm{ho}}
\end{aligned}
\tag{8}
$$

These steps have to be repeated until the multivariate distribution of $\mathbf{X}_{[j+1]}^{\mathrm{hs}}$ matches $\mathbf{X}^{\mathrm{ho}}$, or the similarity between the two matrices cannot be increased further. This is measured using the (squared) energy distance (Székely and Rizzo, 2004, 2013;





Rizzo and Székely, 2016), a measure of statistical discrepancy between two multivariate distributions. For two $N$-dimensional independent random vectors $\mathbf{x}$ and $\mathbf{y}$ with respective CDFs $F$ and $G$, this measure is given by:

$$D^2\left(F,G\right) = 2E\left[|| \ \mathbf{x} - \mathbf{y} \ |||\right] - E\left[|| \ \mathbf{x} - \mathbf{x}' \ |||\right] - E\left[|| \ \mathbf{y} - \mathbf{y}' \ |||\right] \geq 0 \tag{9}$$

with $E$ the expected value, $|| \ . \ ||$ the Euclidean norm and $\mathbf{x}'$ and $\mathbf{x}$ and $\mathbf{y}'$ and $\mathbf{y}$ i.i.d.. A practical way of calculating this measure is as follows (Székely and Rizzo, 2013), with $\mathbf{X} = (X_1, ..., X_{n_1})$ and $\mathbf{Y} = (Y_1, ...Y_{n_2})$:

$$
\begin{aligned}
D\left(\mathbf{X}, \mathbf{Y}\right) = & \frac{2}{n_1 n_2} \sum_{i=1}^{n_1} \sum_{m=1}^{n_2} | \ X_i - Y_m \ | \\
& - \frac{1}{n_1^2} \sum_{i=1}^{n_1} \sum_{j=1}^{n_1} | \ X_i - X_j \ | \\
& - \frac{1}{n_2^2} \sum_{l=1}^{n_2} \sum_{m=1}^{n_2} | \ Y_l - Y_m \ |,
\end{aligned}
\tag{10}
$$

with $i$, $j$, $l$ and $m$ denoting the time steps. In this study, a tolerance of 0.0001 was used to ensure that the computation would not take too long.

As a last step, the preservation of trends of QDM has to be combined with the restoration of the multivariate ranks by the transformations. To do this, first either the equidistant or equiratio version of quantile delta mapping (depending on the variable)

has to be applied to each variable of the original data set $\mathbf{X}^{\mathrm{fs}}$, using $\mathbf{X}^{\mathrm{hs}}$ and $\mathbf{X}^{\mathrm{ho}}$ as historical baseline data. As a final step, the elements of each column of $\mathbf{X}^{\mathrm{fa}}$ are reordered following a method known as the 'Schaake Shuffle' (Clark et al., 2004; Vrac and Friederichs, 2015). In this method, the ranks of each variable of $\mathbf{X}^{\mathrm{fa}}$ are swapped with the ranks of the corresponding variables of $\mathbf{X}^{\mathrm{fs}}_{[j+1]}$, thus reordering the time series' structure according to the ranks of the observations. The Schaake Shuffle can be mathematically formulated as follows (Clark et al., 2004). Let $\mathbf{X}$ be a vector of $n$ time steps of a variable, and $\chi$ be the

sorted vector of $\mathbf{X}$, that is:

$$\mathbf{X} = (x_1, x_2, ...x_n), \ \text{and} \tag{11}$$

$$\chi = \left(x_{(1)}, x_{(2)}, ...x_{(n)}\right), \quad x_{(1)} \leq x_{(2)} ... \leq x_{(n)} \tag{12}$$

Let $\mathbf{Y}$ be a vector of $n$ historical observations and $\gamma$ be the sorted vector of $\mathbf{Y}$:

$$\mathbf{Y} = (y_1, y_2, ...y_n), \ \text{and} \tag{13}$$

$$\gamma = \left(y_{(1)}, y_{(2)}, ...y_{(n)}\right), \quad y_{(1)} \leq y_{(2)} ... \leq y_{(n)}. \tag{14}$$

$\mathbf{B}$ is then the vector of indices describing the original observation number that corresponds to the values in the ordered vector $\gamma$. The main step of the Schaake Shuffle is to reconstruct the reordered vector $\mathbf{X}^{\mathrm{ss}}$:

$$\mathbf{X}^{\mathrm{ss}} = (x_1^{\mathrm{ss}}, x_2^{\mathrm{ss}}, ..., x_n^{\mathrm{ss}}) \tag{15}$$



where

$\quad x_q^{\text{ss}} = x_{(r)},\, q = \mathbf{B}\,[r]\,,\text{and}\, r = 1,...,n$ (16)

This method is essentially the only difference with the univariate QDM, implying that differences in performance can be related to it.

The MBCn method was shown by Cannon (2018) to outperform many earlier multivariate bias-adjusting methods, such as the EC-BC (Vrac and Friederichs, 2015), the JBC (Li et al., 2014), MBCr and MBCp methods (Cannon, 2016). In contrast,
Cannon (2018) also pointed out some problems. Depending on the number of variables, the computational cost can get too high and convergence speed too low, or overfitting might become an issue. The first problem can be tackled by implementing sufficient time steps when having a lot of variables. Second, to address the convergence speed, Pitié et al. (2007) suggested using a deterministic selection of rotation matrices that maximizes the distance between rotation axis sets instead of randomly generating them. It is also suggested by Cannon (2018) to use the most efficient form of quantile mapping and to limit the use
of an advanced quantile mapping technique to the last step. Third, to avoid overfitting (and to reduce the computational cost), early stopping is suggested (e.g. Prechelt (1998)).

### 3.6 Evaluation

To compare and evaluate different bias-adjusting methods, a logical evaluation structure is required. One of the goals of this study is to compare the bias-adjusting methods in a climate change context. To do this, 1970-1989 was chosen as the control
or 'historical' time period and 1998-2017 as the validation or 'future' period. Choosing these time frames allowed for a comparison with observations in the validation period for a time period that was already affected by climate change (IPCC, 2013). For a robust calculation of the bias adjustment, 30 years of data are advised (Berg et al., 2012; Reiter et al., 2018). This decreases the effect of internal variability of the climate model and therefore decreases the uncertainty of the bias adjustment results (Maraun, 2012). However, 30 years of data would in this specific case have resulted in overlapping time series, as no
earlier data are available. This would have consequently decreased the differences between the time series. The 20 years chosen as an alternative were used in a set-up called a 'pseudo-projection' (e.g. Li et al. (2010)). This evaluation set-up resembles the Differential Split-Sample Testing (DSST) implemented by Teutschbein and Seibert (2013), which is based on the work of Klemeš (1986). In DSST, the calibration and validation time series are chosen to make the difference between both as large as possible, which allows to study how robust a model is to changes.
Besides the choice of calibration and validation years, it is important to define the indices used for the evaluation (Table 1). As discussed by Maraun et al. (2015) and Maraun and Widmann (2018), it is important to use indices that are both directly and indirectly affected by the bias adjustment. They argue that only using directly adjusted indices can possibly be misleading, especially when using cross-validation. To account for this, indices based on the discharge $Q$ were used (indices 2 and 3). These indices allowed for the assessment of the impact of the occurrence-bias-adjusting methods on simulated river flow. As a
final important group of indices, occurrence indices were used, in order to assess how the combined methods differ in changing the precipitation occurrence of the time series (indices 4 to 7). For the precipitation amount, the quantiles of the empirical



**Table 1.** Overview of the indices used

| Nr | Index | Name |
|----|-------|------|
| 1 | $P_x$ | Precipitation amount quantiles, with $x$ the quantile considered |
| 2 | $Q_x$ | Discharge quantiles, with $x$ the quantile considered |
| 3 | $Q_{T20}$ | 20-year return period value of discharge |
| 4 | $P_{P00}$ | Precipitation transition probability from a dry to a dry day |
| 5 | $P_{P10}$ | Precipitation transition probability from a wet to a dry day |
| 6 | $N_{dry}$ | Number of dry days |
| 7 | $P_{lag1}$ | Precipitation lag-1 autocorrelation |

distributions were considered. Expert Team on Climate Change Detection and Indices (ETCCDI) precipitation indices (Zhang et al., 2011) have also been considered and calculated. However, these are not included in this paper, as the ETCCDI indices did not allow to discern between the different combined methods. All indices were calculated taking all days into account, instead of only calculating them on wet days.

Similar to Pham et al. (2018), we use the Probability Distributed Model (PDM, Moore (2007)), a lumped conceptual rainfall-runoff model to calculate the discharge for the Grote Nete watershed in Belgium. The PDM as used here was calibrated by Cabus (2008) using the Particle Swarm Optimization algorithm (PSO, Eberhart and Kennedy (1995)). As in Pham et al. (2018), it was assumed that the differences between meteorological conditions in the Grote Nete-watershed and Uccle are negligible, and that thus the adjusted data for the Uccle grid cell can be used as a forcing for the PDM. Furthermore, the goal is not to

make predictions, but to assess the impact of different post-processing chains on the discharge values. To calculate the bias on the indices, both the raw and adjusted RCM time series were used as forcing for this model.

### 3.6.1    Calculation set-up

As some of the bias-adjusting methods include randomness, it is important to have enough computer simulations to account

for variability. Yet, to ensure the availability of sufficient computing capacity, just 20 calculations were carried out for each stochastic step. Only 1 calculation is used in the case of a deterministic method, as more simulations will always result in the same outcome. All methods used here, except for SSR and TDA, are deterministic, thus only the combinations of these methods with respectively QDM and MBCn were simulated 20 times. After these multiple calculations were carried out, the value of each index was first calculated on the basis of every simulation. Then, the values were averaged over the simulations;

these averaged values were used for the comparison of the methods. Biases on the indices are always calculated as simulations minus observations, indicating a positive bias if the simulations are larger than the observations and vice versa.





## 4 Results

The results are shown and discussed according to the used uni- or multivariate intensity methods. For each intensity method, the results are shown for three coherent groups of indices used: precipitation amount, precipitation occurrence and discharge.

To summarize and compare the performance of the methods for each of the index groups, the residual biases for the indices of each combination of occurrence-bias-adjusting and intensity-bias-adjusting methods were calculated, based on the 'added value' concept (discussed in Di Luca et al. (2015)). This residual bias can be calculated relative to the model bias or the observations. This enables a comparison based on how well the methods perform in removing the bias and the size of the bias removal in comparison with the original value for the corresponding index for the observation time series. The residual bias

relative to the observations $\text{RB}_O$ for an index $k$ is calculated as follows

$$\text{RB}_{O\,k} = 1 - \frac{|\text{bias}_{\text{raw},k}| - |\text{bias}_{\text{adj},k}|}{|\text{obs}_k|} \tag{17}$$

wit $\text{obs}_k$, $\text{bias}_{\text{raw},k}$ and $\text{bias}_{\text{adj},k}$ respectively the value of the observations for index $k$, the bias of the raw climate model simulations and of the adjusted climate simulations for index $k$. .

The residual bias relative to the model bias $\text{RB}_{\text{MB}}$ for an index $k$ is calculated as follows

$$\text{RB}_{\text{MB}\,k} = 1 - \frac{|\text{bias}_{\text{raw},k}| - |\text{bias}_{\text{adj},k}|}{|\text{bias}_{\text{raw},k}|} \tag{18}$$

The absolute values are used for calculation to indicate the distance between the raw and adjusted values, thus neglecting if the sign of the bias is changed. If the values of these residual biases are lower than 1 for an index, the method performs better than the raw RCM for this index. For the $\text{RB}_{\text{MB}}$ it is impossible to have values lower than 0. However, for $\text{RB}_O$, this is possible, if the value after bias adjustment is exceptionally small in comparison to the value of the observations. The best methods have

low scores on both residual biases for their indices.

     The original data used for these residual biases, i.e. the observational data, the raw climate simulations and the effective biases of the adjusted simulations, can be found in Appendix A.

### 4.1 Univariate bias adjustment

For the univariate case, QDM was used without a specific occurrence correction, and in combination with thresholding, SSR

and TDA. This yields four time series, which will be compared on basis of the precipitation amount, precipitation occurrence and discharge indices.

#### 4.1.1 Precipitation amount

In Fig. 3, it can be seen that for all occurrence-bias-adjusting methods, the adjusted simulations perform better than the raw climate simulations, for most quantiles, with values ranging from 0.39 to 0.98 for $\text{RB}_O$ and from 0.10 to 0.79 for $\text{RB}_{\text{MB}}$. Only





the highest quantile (99.5) performs worse than the climate simulations and can thus not be plotted. This quantile has for all occurrence methods a value of 1.09 for $RB_O$ and of 2.69 for $RB_{MB}$. This is as expected: the QDM method has been proven in other papers to generally perform well (Cannon et al., 2015). However, in contrast to many other quantile mapping methods (e.g. Li et al. (2010)), quantile delta mapping does not apply extrapolation for the highest quantiles. This extrapolation is often advised for these quantiles, as their simulated values might be larger than the largest values of the observations. Applying no

extrapolation causes underestimation, which is demonstrated here. This implies that QDM might perform even better if this extrapolation or a parametric approach were implemented, as was the case in the original version of Li et al. (2010). This effect is also visible in the other quantiles: the values for both $RB_O$ and $RB_{MB}$ rise with the higher quantiles. This implies that it is harder to correctly adjust the bias for the higher quantiles and that the 90th quantile and higher quantiles are also influenced by this extrapolation problem. Three other quantiles cannot be plotted besides the 99.5th: the 5th, the 25th and 50th quantile. The

value for the observations of the former two is 0 mm, which causes the $RB_0$ to be impossible to calculate. The latter is very small for $RB_O$, caused by a decrease in bias that is larger than the observed value.

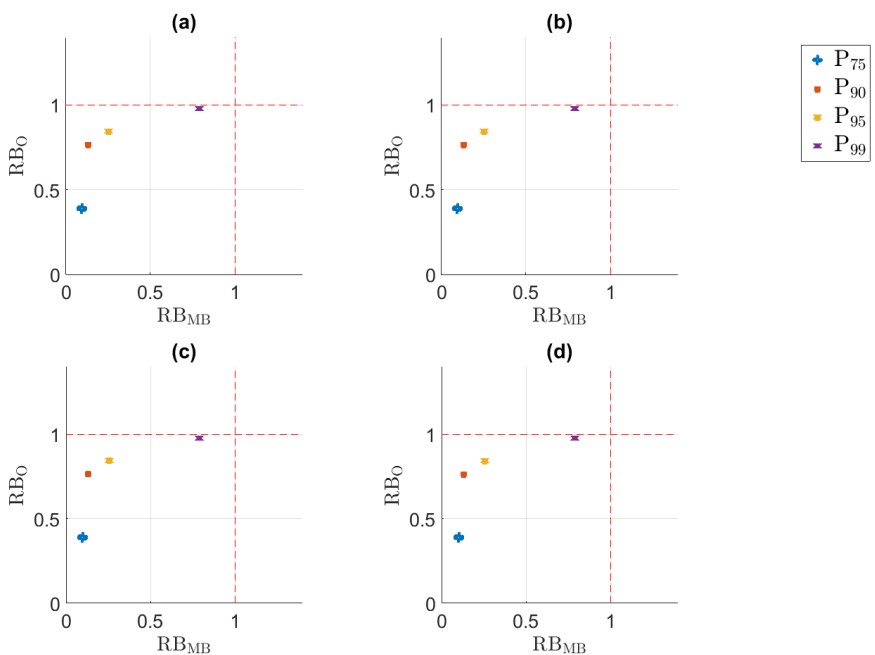

**Figure 3.** $RB_{MB}$ versus $RB_0$ for the univariate precipitation amount quantiles. (a) without specific adjustment, (b) thresholding, (c) SSR, (d) TDA.

As for the comparison of the occurrence-bias-adjusting methods, all methods perform similarly. This was to be expected, as the difference in dry days between the methods is relatively small compared to the total number of dry days and its influence is thus negligible.





### 4.1.2 Precipitation occurrence

For the precipitation occurrence, the indices indicate that all bias-adjusted simulations perform better than the climate simulations (Fig. 4). The values for $RB_0$ range from 0.54 to 0.88 and from 0 to 0.46 for $RB_{MB}$. The $RB_{MB}$ values indicate that the bias is almost entirely removed, although the $RB_0$ values imply that this bias removal does not add much value in comparison to the observations.

In contrast to the precipitation amount indices, the performances of the transition probabilities depend on the occurrence-bias-adjusting method used. The best values are obtained without a specific occurrence-bias-adjusting method or when the thresholding method is used. The SSR method performs the worst of all methods considered, while the TDA has an intermediate performance. An effect by the occurrence-bias-adjusting methods on the transition probabilities was expected by construction. However, the differences noticed here are not in line with the expectations. It is often assumed that additional randomness, which is introduced by the TDA and SSR methods, brings the simulations closer to the observations. Our results seem to contradict this assumption, as TDA and SSR have a worse performance than thresholding or the baseline without any specific occurrence-bias-adjustment. This indicates that the climate models simulated the wet and dry days at the right positions in the time series. In that case, the additional randomness induced on wet days by TDA and SSR might change the simulated time series too much. This is especially striking for the number of dry days, which SSR adjusts worst. Whereas the other methods completely remove the bias in number of dry days, the simulations are still biased after occurrence-bias-adjustment by SSR. This implies that the final step of SSR, the reintroduction of dry days, reintroduces too few dry days.

### 4.1.3 Discharge

For the discharge, the quantiles and the 20-year return period value all indicate a better performance of the adjusted climate simulations than the original climate simulations, as summarized in Fig. 5. The values for $RB_0$ range from 0.04 to 0.74 and from 0.01 to 0.34 for $RB_{MB}$. This is in line with the precipitation amount quantile results mentioned above and implies that adjusted simulations are almost bias-free. The range in values for $RB_0$ is mostly related to the small absolute range of the bias values, ranging from -0.03 m$^3$/s to 0.97 m$^3$/s.

There are some small differences between the occurrence-bias-adjusting methods, indicating that the combination of the non-linear PDM and the occurrence-bias-adjusting methods increases the effect of the latter. However, these differences are subtle and it is hard to discern the best method. Only the 99th quantile for the TDA method indicates a slightly better performance than the other methods. However, this difference is small: TDA has a $RB_{MB}$ value of 0.03 for this quantile whereas the other methods have values of 0.08 or 0.09.

The small, almost indiscernible differences seem to imply that for the combinations of QDM and occurrence-bias-adjusting methods, the more advanced methods do not bring any added value.





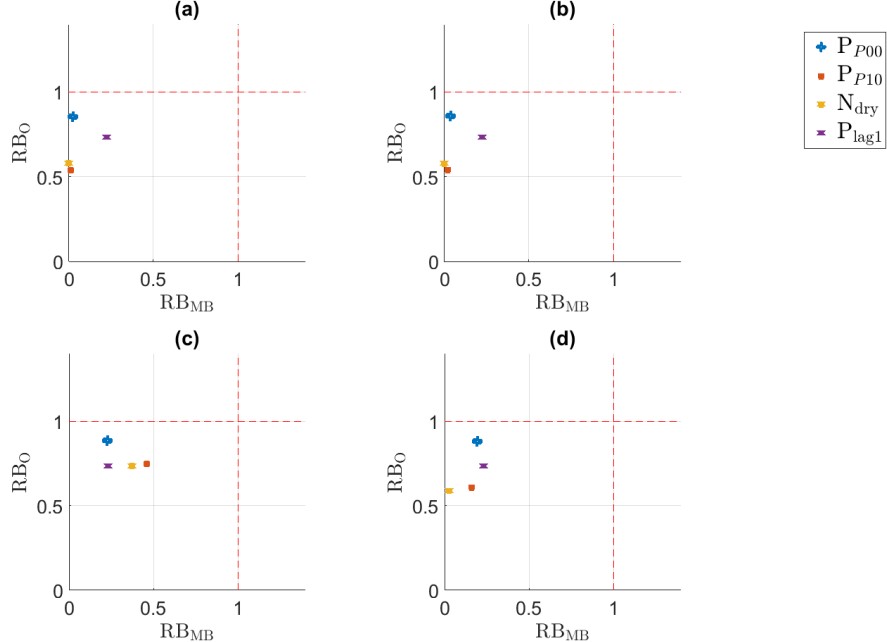

**Figure 4.** $RB_{MB}$ versus $RB_0$ for the univariate precipitation occurrence indices. (a) without specific adjustment, (b) thresholding, (c) SSR, (d) TDA.

## 4.2 Multivariate bias adjustment

For the multivariate case, MBCn was used in combination with no specific occurrence correction, thresholding, SSR and TDA. As in the univariate case, this yields four resulting time series, which will be compared on the basis of the precipitation amount, precipitation occurrence and discharge indices.

### 4.2.1 Precipitation amount

Figure 6 illustrates that the results for the precipitation amount indices of the combinations of occurrence-bias-adjusting methods and MBCn are essentially the same as those for QDM. This follows from the construction of MBCn: in the last steps, QDM is applied and the resulting time series is changed on the basis of the ranks of the previous steps of MBCn, using the Schaake Shuffle. A change in ranks has no influence on the CDF and corresponding quantiles of precipitation amount, which explains that the plots are the same. More information on the specific quantiles can be found above, in Subsection 4.1.1.

### 4.2.2 Precipitation occurrence

Unlike all other index groups, for the combinations with both univariate and multivariate intensity-bias-adjusting methods, the precipitation occurrence indices for the combination of MBCn and occurrence-bias-adjusting methods indicate that the

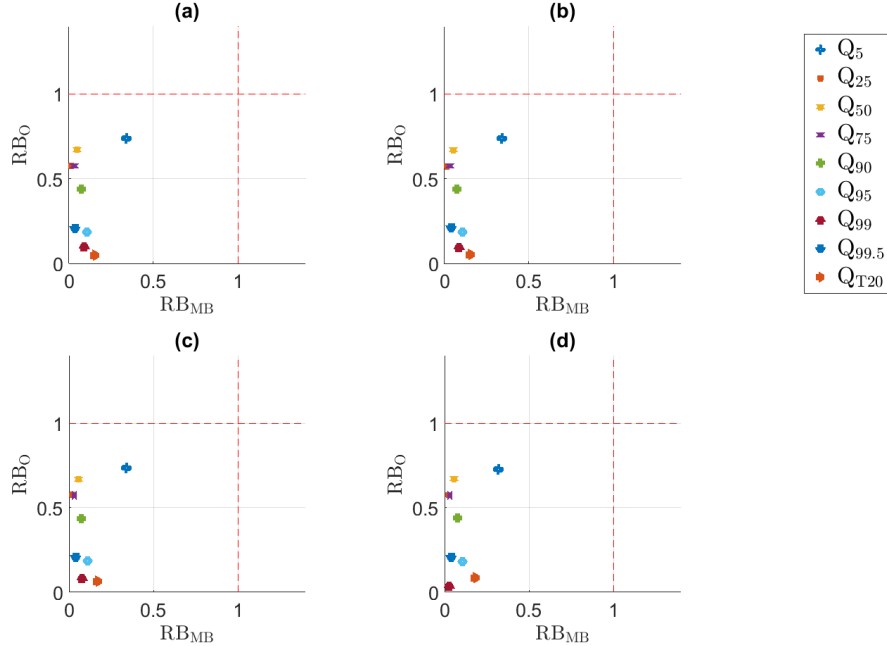

**Figure 5.** $RB_{MB}$ versus $RB_O$ for the univariate discharge quantiles and the 20 year return period value. (a) without specific adjustment, (b) thresholding, (c) SSR, (d) TDA.

adjusted climate simulations generally perform worse than the raw climate simulations (Fig. 7). The wet-to-dry transition probability and precipitation lag-1 autocorrelation are close to the 1-1 line in the plots, but the dry-to-dry transition probability
has $RB_{MB}$ values above 1.5 for every combined method, as can be seen in Table 2. The only two exceptions to the generally poor performance are the wet-to-dry transition probability for SSR and the number of dry days. The former is contrasting with all other index groups, in which the SSR method often performed the worst. However, SSR performed worst for the dry-to-dry transition probability, with values of 1.23 and 2.56 for respectively $RB_O$ and $RB_{MB}$. The number of dry days for the different methods is the same as in the combinations of QDM and the occurrence-bias-adjusting methods. These are unaffected by the
multivariate step of MBCn, as was the case for the precipitation intensity in Section 4.2.1.

Except for SSR, the influence of the occurrence-bias-adjusting methods on precipitation occurrence is negligible, with e.g. values for $P_{P10}$ ranging from 1.03 to 1.04 for $RB_O$ and ranging from 1.07 to 1.09 for $RB_{MB}$. This implies that the performance is largely influenced by the intensity-bias-adjusting method. The only difference between the multivariate and the univariate intensity-bias-adjusting methods is the rank-based Schaake Shuffle, as previously mentioned, which implies that
this shuffling might be responsible for the performance. As the shuffle changes the time series structure, it is a plausible cause for the poor performance of occurrence-related indices. The post-processing done by the SSR method might then balance this effect by reintroducing some dry days, thus increasing the wet-to-dry transition probability. This is in contrast with the other



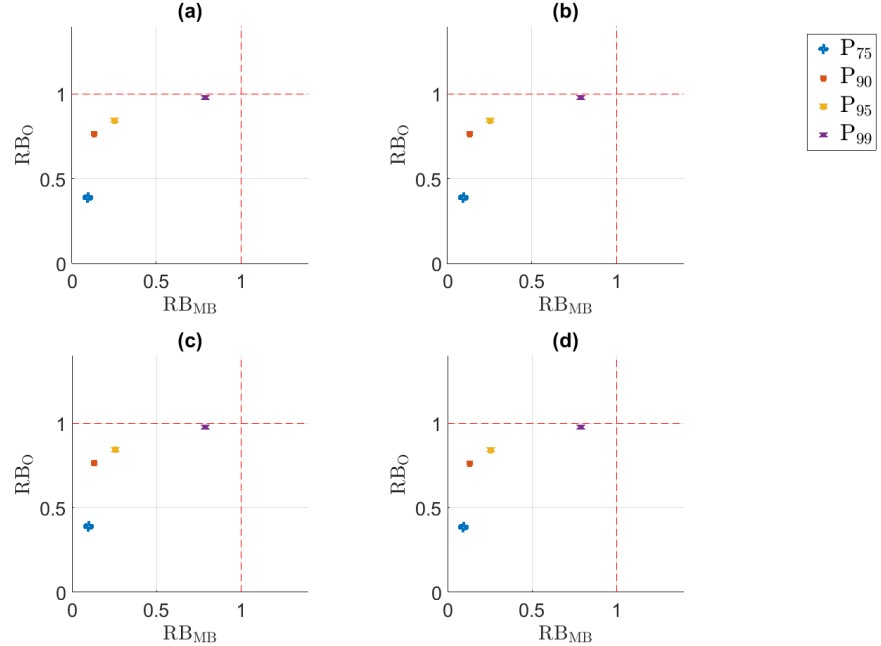

**Figure 6.** $RB_{MB}$ versus $RB_0$ for the multivariate precipitation amount quantiles. (a) without specific adjustment, (b) thresholding, (c) SSR, (d) TDA.

**Table 2.** $RB_{MB}$ and $RB_O$ values of $P_{P00}$ for the combinations of MBCn and the occurrence-bias-adjusting methods

| Occurrence-bias-adjusting method | $RB_{MB}$ | $RB_O$ |
|---|---|---|
| Without specific adjustment | 1.84 | 1.12 |
| Thresholding | 1.81 | 1.12 |
| SSR | 2.56 | 1.23 |
| TDA | 1.73 | 1.11 |

occurrence-bias-adjusting methods, which do not apply post-processing after the intensity correction step. However, this would also explain the poor performance for $P_{P00}$ for SSR, as the post-processing also changes the dry-to-dry transition probability
by reintroducing dry days. This changes the structure of dry day sequences and thus implies an extra change in the temporal structure on top of the effect of the Schaake Shuffle.

### 4.2.3 Discharge

The results of the adjusted climate simulations for discharge (Fig. 8) are all better than the raw climate simulations. In this case, the values for $RB_0$ range from 0.09 to 0.97 and from 0.03 to 0.91 for $RB_{MB}$. Both indices imply that the added value of the bias





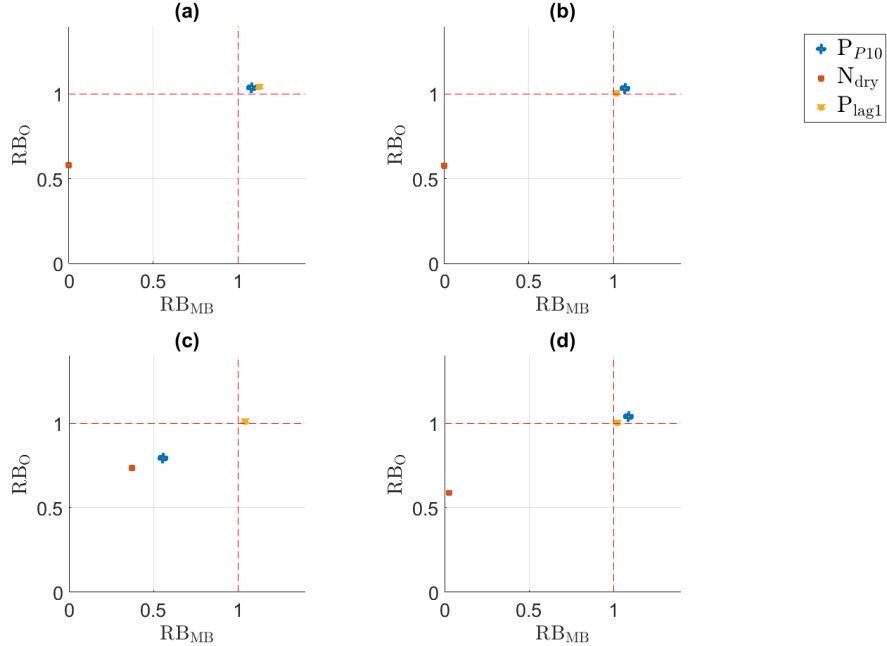

**Figure 7.** $RB_{MB}$ versus $RB_O$ for the multivariate precipitation occurrence indices. (a) without specific adjustment, (b) thresholding, (c) SSR, (d) TDA.

adjustment depends strongly on the quantile under consideration. For the lowest (5th) discharge quantile, the performance of the adjusted climate simulations is only marginally better than for the raw climate simulations. With values ranging from only 0.95 to 0.97 and from 0.97 to 0.91 for respectively $RB_O$ and $RB_{MB}$, it can be assumed that the influence of the occurrence-bias-adjusting methods on the lowest discharge quantiles is negligible. However, the performance of MBCn combined with the occurrence-bias-adjusting methods for the lowest quantile shown here is worse than for the same quantile in Fig. 5. As

mentioned in Section 3.5, the reordering by the Schaake Shuffle is the only difference with QDM. Therefore, it can be assumed that changing the time series structure by the rank-based reordering breaks sequences of low precipitation amounts, which results in less or different low flow sequences. In contrast to the 5th quantile, the resulting performance for the combinations under consideration is better for the highest quantiles, for both $RB_O$ and $RB_{MB}$. This indicates that the climate model bias was large and most of this bias has been removed by the bias-adjustment.

When comparing the occurrence-bias-adjusting methods, variability in their performance can be noticed for the indices under consideration, unlike in Section 4.2.1. However, as can be seen in Table 3, the differences are small for most quantiles. This implies that there is no single method that performs considerably better than any other. Thus for this case, the assumption that randomized steps nudge the adjusted climate simulations closer to the observations does not hold, as was the case for the univariate discharge quantiles in Section 4.1.3.





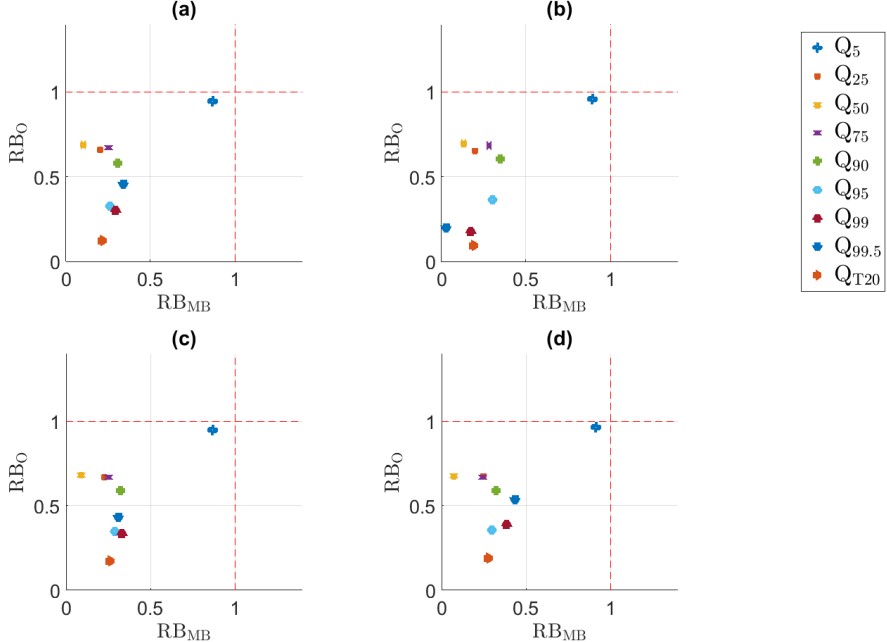

**Figure 8.** $RB_{MB}$ versus $RB_0$ for the multivariate discharge quantiles and 20 year return period value. (a) without specific adjustment, (b) thresholding, (c) SSR, (d) TDA.

**Table 3.** Maximal difference among the discharge indices for the combinations with MBCn for $RB_{MB}$ and $RB_O$

| Index | $RB_{MB}$ | $RB_O$ |
|-------|-----------|--------|
| $Q_5$ | 0.05 | 0.02 |
| $Q_{25}$ | 0.05 | 0.02 |
| $Q_{50}$ | 0.06 | 0.02 |
| $Q_{75}$ | 0.04 | 0.02 |
| $Q_{90}$ | 0.04 | 0.03 |
| $Q_{95}$ | 0.04 | 0.04 |
| $Q_{99}$ | 0.21 | 0.21 |
| $Q_{99.5}$ | 0.41 | 0.34 |
| $Q_{T20}$ | 0.09 | 0.10 |

The range of values is much larger for the highest quantiles, i.e. the 99th and 99.5th quantiles, which contribute to the most extreme events. These quantiles have ranges of respectively 0.21 and 0.34 for $RB_O$ and 0.21 and 0.41 for $RB_{MB}$. It is unexpected that these are influenced by the occurrence-bias-adjusting methods, as these methods are supposed to only change the low precipitation amount values. However, the combination of changes in low precipitation amount values with





the rank-based Schaake Shuffle and the non-linearity of the rainfall-runoff model might cause this effect. This becomes more

clear in comparison with Fig. 5, which includes the same steps, except the Schaake Shuffle based on the preceding rotations. As mentioned above, the highest quantiles are the most sensitive to these structural changes (Switanek et al., 2017). When comparing the CDFs of the combinations under consideration (figure not shown), this sensitivity of the quantiles is visible. This seems to imply that the small effect of the occurrence-bias-adjusting methods (which can be seen in Fig. 4) is reinforced by the multivariate MBCn method. This difference in effect of methods becomes clearer when comparing the highest quantiles

for thresholding with the other occurrence-bias-adjusting methods: the 99.5th quantile performs much better for this method than for the other methods, with values for $RB_O$ and $RB_{MB}$ close to zero. As in Section 4.1.2, the best performing method is not one of the advanced methods, but the simpler thresholding method.

Table A2 shows what the poor performance of the highest quantiles means in practice: the bias in the climate simulation is an overestimation, and whereas all methods correct this overestimation, only the thresholding method does correct too much,

implying that its temporal structure is the closest to the observations and that the temporal structure of the other methods does not allow the precipitation to build up the antecedent conditions for extreme floods. When comparing the time series of the adjusted climate simulations (figures not shown), it can be seen that the extreme floods in time series adjusted by combinations of QDM and occurrence-bias-adjusting methods are situated in periods with strong antecedent conditions, i.e. more than one week of daily rainfall and can be caused by moderate rainfall peaks on top of these antecedent conditions. In contrast, in time

series adjusted by combinations of MBCn and occurrence-bias-adjusting methods, the antecedent conditions are less strong, with only a few days of rainfall or even dry days shortly before the flood event: the floods are caused by extreme rainfall peaks. The combination of MBCn with the thresholding occurrence-bias-adjusting method is situated somewhat in between, causing extreme floods in periods of moderate-to-strong antecedent conditions and extreme rainfall peaks.

## 5   Discussion

When all methods are compared, the performance of the adjusted climate simulations is for the indices considered here generally better than the climate simulations. As a recent time period, 1998-2017, is used for validation purposes, the good performance shows that so far, for precipitation bias adjustment, the stationarity assumption holds and that the adjustments are meaningful in a climate change context. However, this does not give any indication of the time for which this assumption will continue to hold. Only for the precipitation occurrence indices of the combinations of MBCn and the occurrence-bias-adjusting

methods (Fig. 7), the performance is worse than the climate simulations.

When comparing the univariate and multivariate intensity-bias-adjusting methods, some major differences can be noticed. For QDM, the effect of the occurrence-bias-adjusting methods was only seen on the precipitation occurrence indices, which was to be expected. In contrast, for the combinations of the multivariate intensity-bias-adjusting methods and the occurrence-bias-adjusting methods, there were differences in both the discharge and precipitation occurrence indices. These differences

were most pronounced for the methods that apply randomization: they performed unexpectedly worse (Figs. 4 and 8). This is in contrast with the findings of Vrac et al. (2016), who demonstrated the superior performance of SSR in comparison with





other methods. However, Vrac et al. (2016) used a different combination of occurrence and intensity-bias-adjusting methods. First, the occurrence-bias-adjusting methods that were proven to be outperformed by SSR by Vrac et al. (2016), the positive correction and threshold adaptation, were not used in this paper. Second, another univariate method, CDF-t (Michelangeli et al., 2009) was used. Third, no multivariate method was used in the comparison of Vrac et al. (2016). Besides, the geographical region was different, though comparable, as Vrac et al. (2016) used the complete EURO-CORDEX region.

Nonetheless, in the univariate case, which is most comparable to the study by Vrac et al. (2016), both randomness-inducing methods perform worst. The combinations of QDM without a specific adjustment method and with a simpler method such as thresholding, perform best in the univariate precipitation occurrence case (Fig. 4). In the multivariate discharge case (Fig. 8) thresholding performed especially good. This implies that other multivariate methods, which apply different algorithms to restore the multivariate correlation structure, might have a varying impact on the occurrence correction. This has to be studied in more detail.

The effect of using a multivariate intensity-bias-adjusting methods as such is also interesting. When assessing the precipitation occurrence for the combinations with MBCn, it was noticed that the poor performance seemed to be more related to the use of the rank-based Schaake Shuffle alone, than to the use of occurrence-bias-adjusting methods, as they mostly had a similar performance. This effect could also be noticed for the discharge indices for the combinations with MBCn (Fig. 8), though in those combinations the effect of the occurrence-bias-adjusting methods is slightly larger. However, there is not a lot of information available yet on the effect of multivariate intensity-bias-adjusting methods on precipitation occurrence, though this paper partially fills the gap.

## 6 Conclusions

In this paper, different occurrence-bias-adjusting methods were compared in combination with univariate and multivariate intensity-bias-adjusting methods. The occurrence-bias-adjusting methods compared were thresholding, Stochastic Singularity Removal (SSR) and Triangular Distribution Adjustment (TDA). The intensity-bias-adjusting methods were also run once without a specific adjustment method. The intensity-bias-adjusting methods used were QDM (univariate) and MBCn (multivariate). The combinations of occurrence-bias-adjusting and intensity-bias-adjusting methods were compared for precipitation amount, precipitation occurrence and discharge indices, using the residual bias relative to the model bias and the residual bias relative to the observations.

These comparisons have shown that the performance is very specific depending on the combination of bias-adjusting methods. Two main observations could be made. First, in both the univariate and multivariate combinations the randomness-inducing methods, SSR and TDA, performed generally worse than the more traditional methods such as thresholding. They performed even worse than the baseline where the intensity-bias-adjusting method was applied. Second, the multivariate method has a large influence on the performance of the occurrence-bias-adjusting methods, by changing the time series structure and possibly reinforcing the effect of the occurrence-bias-adjusting methods, worsening the performance of the resulting adjusted climate simulations. In the multivariate case, small changes made by the occurrence-bias-adjusting methods thus seem to have

an unexpectedly large influence. This implies that more research should be done on the interaction of multivariate intensity-bias-adjusting methods and occurrence-bias-adjusting methods, to clarify how combined methods influence the assessment of climate change impact. The effect of multivariate methods themselves on the time series structure is also uncertain, and will be investigated in further research. This is especially important for the assessment of hazards that depend on the time series structure, as is the case in hydrology and agriculture. This also has to be assessed for other geographic/climatic regions, where

precipitation characteristics and in turn the interaction between subsequent bias-adjusting methods may be different. To conclude, as long as the effect of the interactions between bias-adjusting methods is unclear, the use of simpler methods seems recommended to reduce the uncertainty as much as possible when assessing and communicating global change impacts. In the framework of this study, the simpler method would be the thresholding occurrence-bias-adjusting method.

*Code and data availability.* The code for the calculcations is publicly available at https://doi.org/10.5281/zenodo.3675305 (Van de Velde,

2020). The RCA4 data are downloaded and are available from the Earth System Grid Federation data repository. The local observations were obtained from RMI in Belgium, and cannot be shared with third parties.

*Author contributions.* JV, BDB and NV designed the experiments. JV developed the code and performed the simulations. JV, BDB and NV designed the visualization. JV prepared the manuscript with contributions from MD, BDB and NV. All co-authors contributed to the interpretation of the results.

*Competing interests.* The authors declare that they have no conflict of interests.

*Acknowledgements.* The authors are grateful to the RMI for allowing the use of 117-year Uccle dataset. This work was funded by FWO, grant number G.0039.18N.





# Appendix A: Appendix A

**Table A1.** Observed values, and biases for the raw climate simulations, and the combinations of QDM and occurrence-bias-adjusting methods.

| Index | Observed value | Bias | | | | |
|---|---|---|---|---|---|---|
| | | Raw climate simulations | Without specific adjustment | SSR | TDA | Thresholding |
| $Q_5$ (m³/s) | 2.30 | 0.92 | -0.31 | -0.31 | -0.29 | -0.31 |
| $Q_{25}$ (m³/s) | 3.36 | 1.45 | 0.02 | 0.03 | 0.03 | 0.02 |
| $Q_{50}$ (m³/s) | 4.39 | 1.53 | 0.08 | 0.08 | 0.09 | 0.08 |
| $Q_{75}$ (m³/s) | 5.72 | 2.52 | -0.08 | -0.08 | -0.08 | -0.08 |
| $Q_{90}$ (m³/s) | 7.83 | 4.76 | -0.36 | -0.35 | -0.37 | -0.36 |
| $Q_{95}$ (m³/s) | 10.09 | 9.22 | -1.01 | -1.00 | -0.97 | -1.00 |
| $Q_{99}$ (m³/s) | 18.71 | 18.58 | -1.67 | -1.41 | -0.53 | -1.65 |
| $Q_{99.5}$ (m³/s) | 23.90 | 19.70 | -0.77 | 0.78 | 0.74 | 0.84 |
| $Q_{T20}$ (m³/s) | 48.69 | 54.61 | 8.30 | 9.13 | 9.99 | 8.36 |
| $P_5$ (mm) | 0.00 | 0.00 | 0.00 | 0.00 | 0.00 | 0.00 |
| $P_{25}$ (mm) | 0.00 | 0.08 | 0.00 | 0.00 | 0.02 | 0.00 |
| $P_{50}$ (mm) | 0.10 | 1.01 | 0.05 | 0.06 | 0.04 | 0.05 |
| $P_{75}$ (mm) | 2.70 | 1.83 | -0.18 | -0.18 | -0.19 | -0.18 |
| $P_{90}$ (mm) | 7.40 | 1.99 | -0.26 | -0.26 | -0.26 | -0.26 |
| $P_{95}$ (mm) | 11.42 | 2.38 | -0.61 | -0.61 | -0.61 | -0.61 |
| $P_{99}$ (mm) | 21.80 | 2.36 | -1.86 | -1.86 | -1.86 | -1.86 |
| $P_{99.5}$ (mm) | 29.09 | 1.56 | -4.20 | -4.20 | -4.20 | -4.20 |
| $P_{P00}$ | 0.65 | -0.10 | 0.00 | -0.02 | -0.02 | 0.00 |
| $P_{P10}$ | 0.32 | -0.15 | 0.00 | -0.07 | 0.02 | 0.00 |
| $N_{dry}$ | 3470.00 | -1466.00 | 0.00 | -544.70 | 38.00 | 0.00 |
| $P_{lag1}$ | 0.33 | 0.11 | 0.03 | 0.03 | 0.03 | 0.03 |





**Table A2.** Observed values, and biases for the raw climate simulations, and the combinations of MBCn and occurrence-bias-adjusting methods.

| Index | Observed values | Bias | | | | |
|---|---|---|---|---|---|---|
| | | Raw climate simulations | Without specific adjustment | SSR | TDA | Thresholding |
| $Q_5$ (m$^3$/s) | 2.30 | 0.92 | 0.79 | 0.80 | 0.84 | 0.82 |
| $Q_{25}$ (m$^3$/s) | 3.36 | 1.45 | 0.29 | 0.33 | 0.36 | 0.29 |
| $Q_{50}$ (m$^3$/s) | 4.39 | 1.53 | -0.16 | -0.14 | -0.11 | -0.20 |
| $Q_{75}$ (m$^3$/s) | 5.72 | 2.52 | -0.64 | -0.63 | -0.62 | -0.72 |
| $Q_{90}$ (m$^3$/s) | 7.83 | 4.76 | -1.46 | -1.54 | -1.55 | -1.66 |
| $Q_{95}$ (m$^3$/s) | 10.09 | 9.22 | -2.42 | -2.66 | -2.74 | -2.78 |
| $Q_{99}$ (m$^3$/s) | 18.71 | 18.58 | -5.51 | -6.17 | -7.17 | -3.21 |
| $Q_{99.5}$ (m$^3$/s) | 23.90 | 19.70 | -6.72 | -6.15 | -8.58 | -0.57 |
| $Q_{T20}$ (m$^3$/s) | 48.69 | 54.61 | -11.85 | -14.26 | -15.16 | -10.40 |
| $P_5$ (mm) | 0.00 | 0.00 | 0.00 | 0.00 | 0.00 | 0.00 |
| $P_{25}$ (mm) | 0.00 | 0.08 | 0.00 | 0.00 | 0.02 | 0.00 |
| $P_{50}$ (mm) | 0.10 | 1.01 | 0.05 | 0.06 | 0.04 | 0.05 |
| $P_{75}$ (mm) | 2.70 | 1.83 | -0.18 | -0.18 | -0.19 | -0.18 |
| $P_{90}$ (mm) | 7.40 | 1.99 | -0.26 | -0.26 | -0.26 | -0.26 |
| $P_{95}$ (mm) | 11.42 | 2.38 | -0.61 | -0.61 | -0.61 | -0.61 |
| $P_{99}$ (mm) | 21.80 | 2.36 | -1.86 | -1.86 | -1.86 | -1.86 |
| $P_{99.5}$ (mm) | 29.09 | 1.56 | -4.20 | -4.20 | -4.20 | -4.20 |
| $P_{P00}$ | 0.65 | -0.10 | -0.18 | -0.25 | -0.17 | -0.17 |
| $P_{P10}$ | 0.32 | -0.15 | 0.16 | 0.08 | 0.16 | 0.16 |
| $N_{dry}$ | 3470.00 | -1466.00 | 0.00 | -544.70 | 41.15 | 0.00 |
| $P_{lag1}$ | 0.33 | 0.11 | -0.13 | -0.12 | -0.12 | -0.12 |



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
