# Peer review of "Comparison of occurrence-bias-adjusting methods for hydrological impact modelling"

_Hydrology and Earth System Sciences, 2020_

## Short Comment (SC1) · 29 Apr 2020

The issue of precipitation occurrence in a bias adjustment context is interesting and deserves more attention. Without yet having read the manuscript in great detail, I find the study well conducted and useful. I just want to let you know that we proposed a simple but practical empirical approach to tackle this problem in Olsson et al. (2012), in the context of Delta Change and high-resolution rainfall time series. Please consider adding this study in the literature review of previous work on this topic.

Olsson, J., Gidhagen, L., Gamerith, V., Gruber, G., Hoppe, H., and P. Kutschera (2012) Downscaling of short-term precipitation from Regional Climate Models for sustainable

urban planning, Sustainability, 4, 866-887, doi:10.3390/su4050866.

---

## Short Comment (SC2) · 1 Jul 2020

As a person who is currently working on multivariate bias correction, I really enjoyed reading this manuscript. Although the overall view of jointly bias correcting variables seemed quite promising since a decade ago, the uncertainties that come with adopting complex methods can not be taken for granted. Therefore, multivariate bias correction needs further investigation and I personally find your manuscript a good initiative in assessment of these methods. I think you made a good point by your conclusion: "The use of simpler methods seems recommended to reduce the uncertainty as much as possible when assessing and communicating global change impacts."

[Figure]

For the sake of clarity, I'd like to make some comments on the text; pointing out phrases that I couldn't understand thoroughly (though I am not the person who is officially assigned to do that).

Line 62-63: such as CDF-t (Michelangeli et al. (2009), used in e.g. Vrac (2018)) : -Maybe better to change to: such as CDF-t which was originally proposed by Michelangeli et al. (2009) and later used in e.g. Vrac (2018).

Line 154-155: Third, both historical and future simulations are adjusted at the same moment, to ensure a sound comparison during the intensity phase of the adjustment: -I do not fully understand what do you mean by moment? Do you mean mathematical moments or time? And if it is the latter, how does it insure a sound comparison during the intensity phase?

Line 159: To overcome the assumption that the simulated time series has to have more wet days than the observations... -Why simulated time series must have more wet days than observations?

Line 286: This method is essentially the only difference with the univariate QDM, implying that differences in performance can be related to it. -This sentence is a bit unclear to me.

Line 332-333: Thus, only the combinations of these methods with respectively (respective?) QDM and MBCn were simulated 20 times. -This sentence is a bit unclear to me.

Line 368-369: "This" extrapolation is often advised for these quantiles, as their simulated values might be larger than the largest values of the observations. -It is a bit unclear to me what is "this" referring to.

Line 484: When all methods are compared, the performance of the adjusted climate simulations is for the indices considered here generally better than the climate simulations. -This sentence is a bit unclear to me.

---

## Referee Comment (RC1) · Stefan Lange (Referee) · 22 Jul 2020

**General comments**

The authors of "Comparison of occurrence-bias-adjusting methods for hydrological impact modelling" test different methods for the bias adjustment (BA) of daily precipitation time series. In particular, they combine different methods for dry-day frequency BA with one method for the BA of wet-day precipitation intensities (Quantile Delta Mapping or QDM), and they test how an additional temporal shuffling of the time series for the purpose of multivariate bias adjustment (MBA) influences the results. The different method combinations are evaluated using precipitation occurrence statistics, precip-

itation quantiles, and discharge quantiles, with discharge computed using a lumped rainfall-runoff model. The authors find that the stochasticity added by the more complicated methods worsens results and conclude that simpler methods should be used for BA until the stochasticity-related problems identified here are better understood.

While I see value in investigating the performance of combined BA methods whose components have only been separately evaluated before, I am afraid that the present study is not a good example of such an investigation, for the following three main reasons. First, QDM alone is found to already perfectly adjust the dry-day frequency, hence no added value can be expected from combining QDM with any designated dry-day frequency BA method. Secondly, most of those combinations are done such that they are bound to fail by construction. Thirdly, where bad performance is found it is not explained mechanistically.

The first point is so important because the biggest differences in performance between method combinations were found for precipitation occurrence statistics. The authors write that QDM is only applied to wet-day precipitation (more than 0.1 mm/day), yet for some mysterious reason that application also perfectly adjusts the dry-day frequency (Fig. 4, Table A1). This result, which surprises me, is neither discussed nor explained. It could be related to a QDM application that was not done as described. Or it happened by chance and would not happen again if different climate data were used. In any case, it is an unfortunate result because it has the effect that no other method combination was able to better adjust the dry-day frequency and hence the simplest approach was bound to perform best.

The second point makes that even worse. You combine Stochastic Singularity Removal (SSR) and Triangular Distribution Adjustment (TDA) with QDM such that they are unable to adjust the dry-day frequency by construction. Your version of SSR+QDM first turns all dry days into wet days; then QDM is applied to all days (because they are all wet now), which changes the number of days with precipitation less than $P_{\min}$ in an uncontrolled way; then all days with precipitation less than $P_{\min}$ are turned into dry

days, so the dry-day frequency is adjusted in an uncontrolled way. The problem is less pronounced but similar for TDA+QDM, where TDA is first applied to properly adjust the dry-day frequency, but then QDM is applied, which moves some formerly wet days below the wet-day threshold and hence turns them into dry days; therefore TDA+QDM always generates too many dry days (Tables A1 and A2).

The third point is that as you can read above it is relatively easy to explain most of your results. Yet you do not even try to do that. Just presenting the results without explaining them will not help others to better combine bias adjustment methods in the future.

**Specific comments**

l 157: I think you should mention that your Thresholding method is not able to account for a climate change signal in the dry-day frequency.

l 165-167: Vrac et al. (2016) set $P_{\min} = 8.64\,10^{-4}$ mm/day. That is much lower than your $P_{\min}$ value and I am sure that this influences your results.

l 173: Why is SSR not also applied month by month like Thresholding and TDA? Say so if that would not make a difference.

l 175: What is the motivation behind TDA? You only explain how the method works but not why (with which intentions) it was developed. What can it do better than other methods?

l 177f: What if that is not possible, for example if $f^{\text{ho}} = 0.9$, $f^{\text{hs}} = 0.3$, $f^{\text{fs}} = 0.5$?

l 194: Not clear to me: Do you loop through 1–3 one day at a time or how? I am confused because l 189f suggests something else.

Fig. 2: This figure is currently not helping to understand the TDA method. Please add text to the caption that explains what you intend to illustrate here. What are the red and black arrows? Is this supposed to explain the dry-to-wet case or the wet-to-dry case? Is precipitation decreasing on the x axis in panel (a) and if yes then why? Why

is $x_{th} < 1mm$, should it not be bigger because $b = 0.9$? Is that supposed to be the CDF of the triangular distribution in panel (b) and if yes then why is the curve not smoother?

l 218-220: There are indices $i$ missing in the innermost brackets of both equations.

l 221f: Which version is used for evaporation?

l 223: You should also specify how many empirical quantiles you used in your QDM configuration.

l 227-229: Which threshold was used for evaporation and why? What do you mean with "calculated"? Do you mean that the empirical CDFs in Eq. (6) where based on wet-day values only? Please be more specific.

l 248-250: Does that mean that MBCn was only applied to days with precipitation > 0.1 mm and evaporation > 0.1 mm? Or do you mean that QDM in the last step of MBCn was only applied to days with precipitation > 0.1 mm and evaporation > 0.1 mm?

l 255f: Cannon (2016) uses the additive version of QDM for that. Did you do the same?

l 270-285: This description is quite confusing. I think you do not need to mention the Schaake shuffle and you do not need Eqs. (11-16) because all you do in the last step is shuffle $\mathbf{X}^{fa}$ along the time axis such that the rank time series become identical to those of $\mathbf{X}^{fs}_{[k]}$, where $k$ is the number of iterations of steps one to three.

l 286f: Does that imply that MBCn was also applied using 91-day running windows etc.? You better be more explicit here.

l 295f: Cannon (2016) also suggests early stopping. Have you tested that? What did you do to detect and prevent overfitting?

l 297: It would help the readers a lot (see below) if you inserted an extra section here that spells out how exactly you combine the dry-day frequency BA methods with QDM/MBCn, and how QDM/MBCn are applied when they are not combined with any dedicated dry-day frequency BA method. For example, in TDA+QDM, is TDA done

before or after QDM? The same for Thresholding+QDM etc.

l 310: You also need to specify whether you account for seasonality or just pool data from all days of the year.

l 312: Why can that be misleading?

l 321-327: Which variables are used to drive PDM? Precipitation only or precipitation, evaporation, and temperature? I assume all three variables because otherwise it would make no sense to apply MBCn.

l 327: What did you use as your true discharge? Did you use observed discharge or did you generate it running PDM based on the observed climate data?

l 332: MBCn is also stochastic, or did you use fixed rotation matrices?

l 368: Which kind of extrapolation are you talking about here? There is no extrapolation necessary to make Eq. (6) work for all values of $x^{\text{fs}}$.

l 380-396: It is impossible to understand these results if you do not spell out exactly how you combine the dry-day frequency BA methods with QDM and how QDM is applied when it is not combined with any dedicated dry-day frequency BA method.

l 389f: Who assumes that? Do you have a reference or can you motivate that assumption? I would not have expected the stochasticity added by SSR and TDA to improve the time series.

l 403-407: Your discussion of those small differences raises the question of statistical significance. There are no uncertainty estimates provided anywhere in your results section. Either you change that (using bootstrapping or the uncertainty originating from the stochastic nature of SSR and TDA, for example) or you are more careful with such statements.

l 421-423: I do not understand. Please rewrite.

[Figure]

l 423-428: This part is quite difficult to read. Please rewrite. You are just describing what can be seen in Fig. 7. This can be done more intelligibly.

l 432: Do those ranges include results for SSR or not?

l 432f: Here you start discussing the influence of QDM on precipitation occurrence but then you just stop and do not follow up in the discussion section.

l 435: "might be responsible for the performance" and "plausible cause": Why so vague? The shuffling changes autocorrelation and transition probabilities but not the distribution. There is no doubt about that.

l 436-438: Aha. You should mention this earlier, ideally in a dedicated section, see above.

Table 2: It is unfortunate that you do not just plot all index values in your figures. I suggest you just extend the axis limits and get rid of this table. Then all the information can be found in one place and in one format.

l 445-448: Can you explain that?

l 450-452: Is that conjecture supported by $Q_5$ being too high?

l 452 You keep writing "5th quantile" where I think you should write "5th percentile".

l 457f: Again, why would you assume that?

Table 3: What is the point of this table? All of this can be seen in Fig. 8, so I suggest to remove it.

l 461-465: I think you are right but this can be spelled out more: SSR and TDA change precipitation ranks randomly and the partly randomized ranks are then shuffled by MBCn. Then PDM calculates discharge, the result of hydrological processes over many days. One could say that discharge integrates precipitation (an other fluxes) over space and time. Therefore, changing the temporal order of events has an effect

on discharge. The impact on the high discharge quantiles is particularly strong for TDA (Fig. 8 vs. Fig. 5) because by setting $b = 0.9$, you have configured TDA such that it may turn pretty heavy precipitation days into dry days.

l 466: Why do you cite Switanek et al. (2017) here? What in that paper is related to your statement?

l 467: I think you should show those figures.

l 469-472: You are talking about quite an extreme quantile here. Such a quantile can only be estimated with limited precision. You really have to quantify the statistical uncertainty of your results and measure the significance of differences if you want to make statements like these.

l 473-476: I am unable to see in Table A2 what you write here.

l 482f: Also about this I would have liked to read a deeper discussion.

l 489-490: That is not correct for all indices.

l 495: You write that SSR+QDM and TDA+QDM performed "unexpectedly" worse. I think they did not, see my general comments.

l 495-501: What you do not mention here is that Vrac et al. (2016) used a much lower $P_{min}$ value (see above) and that they replaced zeros with random numbers from $]0, P_{min}[$. Then they apply CDFt, which by design adjusts the dry-day frequency, in contrast to QDM. The SSR method was designed to work well in combination with CDFt. You are not using SSR as intended by Vrac et al. (2016).

l 506f: This has already been done, see for instance François et al. (2020, doi:10.5194/esd-11-537-2020).

l 524f: That is easily explained by the drizzle effect which is of course perfectly adjusted by thresholding but not by SSR+QDM and TDA+QDM, see my general comments.

l 525f: That needs an explanation!

l 526f: That statement is just wrong. MBCn has no influence on the performance of the dry-day frequency BA methods. All it does is shuffle the time series.

l 530-535: To also say something positive, I support those statements.

l 535-538: I support your call to always use the simplest possible method. Yet as outlined in the general comments, I am afraid your study is ill-designed to support it.

Table A1: QDM alone does not only perfectly adjust $N_{\text{dry}}$ but also $P_{P00}$ and $P_{P10}$. How is that possible?

Tables A1 and A2: I suggest you use better column headings: QDM, SSR+QDM, TDA+QDM, Thresholding+QDM, and similarly for MBCn.

---

## Referee Comment (RC2) · Anonymous Referee #2 · 31 Aug 2020

This paper compares different bias adjustment (BA) schemes for climate data as used in hydrological impact studies. With respect to precipitation, the originality resides in the fact that intensity and occurrence BA methods are used. Also, univariate and multivariate schemes are compared. The analysis is based on a high quality climate data set, yet only a 20 year window is used to make the comparisons. For the comparison, precipitation amount, precipitation occurrence and discharge are evaluated. The results suggest that for the randomness inducing BA methods, performs worse than traditional methods. The results also show that the multi-variate methods lack robustness with respect to the modelling of precipitation occurrence and change the intrinsic structure of the time series in an unexplained way. The results suggest also that the

performance of the current analysis is case specific and lacks generalization (the analysis should be repeated for climate series of other climate regions, and in the case of the multivariate method probably also with other climate variables). The paper is very detailed and uses state-of-the-art methods for implementing BA and for comparing novel BA approaches. The paper is also novel and original as it compares different BA methods for a specific climate region using a high quality data set. Yet the paper lacks focus and is difficult to read. For instance, the BA is on climate data but the authors report also on the impact of climate data on modelled discharge. While this may be interesting to evaluate the impact of BA on final hydrological impact assessments, it does not add value to the comparison of the climate data in-se and only results in less focus of the manuscript. The final conclusions are also somehow disappointing: the addition of complexity in the BA did not result in better results. It is regretted that the reasons for these poor performances of more complex approaches are not explained. The reader is left with a feeling of : "So what? Why do the simple BA approaches outperform as compared to the more complex approaches". No real answers on this question has been offered by the authors.

Small editorials: - Title: The title could be reformulated to focus on the climate data. - It is suggested to create a clear "material and methods" section. - Line 80 – 103. This section should be moved to a "material and methods" section. - Line 105 – 109 is not needed. - Line 310 -314. This section can be removed. - Table 1. Index 2 & 3 could be removed. - Line 321-328 could be removed. - Line 338-357. This section could be moved to a "material and methods section". - Fig.5, Fig. 8. The quality of the symbols should be revised.

---

## Editor Comment (EC1) · Marnik Vanclooster (Editor) · 31 Aug 2020

Dear Authors,

I apologise for the delay in closing the discussion on your paper HESS 2020 83. The paper was send out to many potential reviewers, but who declined systematically the review invitation. This may suggest that there is little scientific interest in the study sofar. Fortunately, 2 referees finally send their report.

The first reviewer suggest to reject the paper as he observes 3 major methodological concerns. The second reviewer suggests a major revision.

I read the mansucript and review reports in detail. I consider the first review as very critical and the concerns raised by this reviewer should be completely addressed in your final answer. In case you consider that these concerns cannot be addressed sufficiently, you can consider to withdraw the paper.

I look forward in learning your final response on these reviewer comments.

Sincerely yours,

---

## Author Comment (AC1) · 7 Oct 2020

**Response to Short comments on 'Comparison of occurrence-bias-adjusting methods for hydrological impact modelling' by J. Van de Velde et al.**

**J. Olsson**

We would like to thank Jonas Olsson for his supportive statements on and his interest in the manuscript. He suggested an additional reference. We added several references at the original l. 48.

*As various other occurrence-bias-adjusting methods are also available and seem to perform well (e.g. Schmidli et al. (2006), Olsson et al. (2012), Ntegeka et al. (2014)), the goal of this paper is to compare a selection of the available methods and study how they interact with the subsequent intensity-bias-adjusting methods. Most of the common bias-adjusting methods, and therefore also the methods used in this comparison, stem from the quantile-mapping-method family (Panofsky et al.,1958).*

**F. Tootoonchi**

We would like to thank Faranak Tootoonchi for her interest in our manuscript and taking the time to read it and give some comments. We will respond comment-per-comment.

**Comment:** l 62-63: Maybe better to change to: such as CDF-t which was originally proposed by Michelangeli et al. (2009) and later used in e.g. Vrac (2018)

**Response:** Thank you for the suggestion, this does increase the readability of the sentence. We have updated the text.

*Older quantile mapping methods are also still in use, such as CDF-t, which was originally proposed by Michelangeli et al. (2009) and later used in e.g. Vrac et a. (2018), and standard empirical quantile mapping, used in e.g. Räty et al. (2018) and Zscheischler et al. (2019).*

**Comment:** L 154-155: I do not fully understand what do you mean by moment? Do you mean mathematical moments or time? And if it is the latter, how does it insure a sound comparison during the intensity phase?

**Response:** What we mean is that both time series are adjusted in the same calculation step (i.e. the moment in time). The sentence has been updated to clarify this.

*Third, both historical and future simulations are adjusted during the same calculation step, to ensure a sound comparison during the intensity phase of the adjustment.*

**Comment:** L 159 Why simulated time series must have more wet days than observations?

**Response:** This is the assumption mentioned in the first paragraph of the previous section (Section 3.1). To clarify this link, the sentence has been expanded:

*To overcome the assumption needed by methods such as thresholding that the simulated time series has to have more wet days than the observations, Vrac et al. (2016) proposed the Stochastic Singularity Removal (SSR) method.*

**Comment:** l 286: This sentence is a bit unclear to me.

**Response:** This sentence was altered in response to the Referee comment by Stefan Lange.

*The version of QDM applied in the step prior to the Schaake Shuffle is the same as in Section 3.4. As such, the shuffling procedure is the only difference with the univariate bias-adjustment, implying that differences in performance can be related to it.*

**Comment:** l 332-333: This sentence is a bit unclear to me.

**Response:** This sentence has been rewritten.

*All methods used here, except for SSR and TDA, are deterministic, thus only the combinations of SSR and TDA with  QDM and MBCn were simulated 20 times.*

**Comment:** l 368-369: It is a bit unclear to me what is "this" referring to.

**Response:** The previous sentence has been altered in response to the referee comment by Stefan Lange. Hopefully this change clarifies the link between the sentences

*However, in contrast to what is advised for many other quantile mapping methods (e.g. Li et al. (2010)), quantile delta mapping does not explicitly apply extrapolation on the observed time series if the highest simulated percentiles are larger than the largest observed value. *

**Comment:** l. 484: This sentence is a bit unclear to me

**Response:** The sentence has rewritten.

*In general, the discharge simulated by the adjusted climate simulations performs better than that simulated by the raw climate simulations.*

---

## Author Comment (AC2) · 7 Oct 2020

**Response to Referee Comment 1 on 'Comparison of occurrence-bias-adjusting methods for hydrological impact modelling' by J. Van de Velde et al.**

**Stefan Lange (Referee)**

We would like to thank Dr. Stefan Lange, referee, for the time spent reviewing our manuscript. Below, we give an overview of his comments, our responses and textual changes. For a detailed in-manuscript overview, we refer to the accompanying file.

**General Comments**

**Comment:** While I see value in investigating the performance of combined BA methods whose components have only been separately evaluated before, I am afraid that the present study is not a good example of such an investigation, for the following three main reasons. First, QDM alone is found to already perfectly adjust the dry-day frequency, hence no added value can be expected from combining QDM with any designated dry day frequency BA method. Secondly, most of those combinations are done such that they are bound to fail by construction. Thirdly, where bad performance is found it is not explained mechanistically.

**Response:** We mostly disagree on the first two points, but agree with the third point. First, we think there is value in showing that QDM alone is enough to adjust the dry-day-frequency, as it is often thought that occurrence-bias-adjusting methods are always needed. However, we did not discuss the consequences of this well enough. Secondly, we are of the opinion that our framework for applying bias adjustment was good and that the failure of methods depended not only on the framework, but also on other circumstances. Yet, we agree that we did not explain the mechanisms clearly enough and thus we hope that the adapted version of the paper now contains sufficient explanation. In general, we understand that this study could have been more extensive, but this was out of scope within the broader setting of our research. With this paper, we wanted to stimulate more discussion on the use of the occurrence-bias-adjusting methods and we hope that we were able to do so and that other researchers are able to expand on these results.

**Comment:** The first point is so important because the biggest differences in performance between method combinations were found for precipitation occurrence statistics. The authors write that QDM is only applied to wet-day precipitation (more than 0.1 mm/day), yet for some mysterious reason that application also perfectly adjusts the dry-day frequency (Fig. 4, Table A1). This result, which surprises me, is neither discussed nor explained. It could be related to a QDM application that was not done as described. Or it happened by chance and would not happen again if different climate data were used. In any case, it is an unfortunate result because it has the effect that no other method combination was able to better adjust the dry-day frequency and hence the simplest approach was bound to perform best.

**Response:** It is correctly pointed out that the QDM application results were not fully described. We have updated this for clarification (see the specific comments regarding this point). However, we fail to see how this would invalidate the remaining results. Instead, it is a result in itself that the baseline (QDM and to a lesser extent also MBCn) also works well. This implies, as has been added in Section 3.1.2, that the simulations are able to correctly simulate the temporal structure, albeitwith a positive bias. That QDM, and implicitly, other quantile mapping-based methods, are thus good enough to adjust the occurrence biases (which are rather intensity biases), is often glossed over in most articles. Mostly, occurrence-bias-adjusting methods are applied without a second thought, or the problem is not mentioned at all. As such, the result for QDM is very interesting and the discussion and conclusion have been expanded accordingly.

(Discussion) *Third, no multivariate method was used in the comparison of Vrac et al. (2016). Besides, the geographical region was different, though comparable, as Vrac et al. (2016) used the complete EURO-CORDEX region. Yet, the most important difference might be the temporal structure of the time series used. The almost perfect adjustment of precipitation occurrence by QDM, i.e. the direct approach, implies that the temporal structure of the time series is simulated well for the region studied here and a mere intensity-bias adjustment is enough to adjust the occurrence biases as well. In contrast, when averaging over a larger area, some regions might have a more poorly simulated temporal structure. In such areas, randomly adjusting the temporal structure might have an added value, as concluded by Vrac et al. (2016). This difference in the best method between both studies is in contrast with the common application of occurrence-bias-adjusting methods. Often, a single occurrence-bias-adjusting method is applied, without considering how it could influence the results and without considering whether or not its application is necessary.*

(Conclusion) *A second question is whether occurrence-bias-adjusting methods are necessary. As indicated by the results of direct approach, this application suffices when the temporal structure of the precipitation time series is simulated good enough, i.e. when the occurrence bias is highly dependent on intensity bias. Thus, it may be relevant to test the performance of a time series adjusted by the direct approach before considering occurrence-bias-adjustment.*

**Comment:** The second point makes that even worse. You combine Stochastic Singularity Removal (SSR) and Triangular Distribution Adjustment (TDA) with QDM such that they are unable to adjust the dry-day frequency by construction. Your version of SSR+QDM first turns all dry days into wet days; then QDM is applied to all days (because they are all wet now), which changes the number of days with precipitation less than Pmin in an uncontrolled way; then all days with precipitation less than Pmin are turned into dry days, so the dry-day frequency is adjusted in an uncontrolled way. The problem is less pronounced but similar for TDA+QDM, where TDA is first applied to properly adjust the dry-day frequency, but then QDM is applied, which moves some formerly wet days below the wet-day threshold and hence turns them into dry days; therefore TDA+QDM always generates too many dry days (Tables A1 and A2).

**Response:** We disagree with this, based on a few points. Firstly, we applied these methods similar to how they, or related methods, are applied in other studies (e.g. Cannon et al. (2015), Vrac et al. (2016), Casanueva et al. (2020)) and as they could easily be applied by practitioners, by first adjusting the occurrence bias and subsequently the intensity bias. Secondly, as Vrac et al. (2016) mentioned, other intensity-bias-adjusting methods could be combined with SSR. Given that SSR was the best method in that paper, it seemed fair to compare it with other methods for occurrence-adjustment and to assume that it would function good as well when combined with SSR. A method very similar to SSR has been applied in combination with QDM in a recent paper by Casanueva et al. (2020), strengthening our opinion. Thirdly, as the framework for applying SSR is thus similar to other papers, we think that how the combinations are applied is not wrong, but an inherent part of their application and thus something that should be discussed. The 'uncontrolled' randomization used by this method and TDA are not necessarily wrong: they allow to create an ensemble of results, which can, especially if the simulated temporal structure is incorrect, result in a better 'ensemble mean'. However, this use of the randomization was not clearly spelled out in the manuscript and has now been updated. Given these points, we'd rather state that, in contrast with the earlier research, SSR and TDA did not perform as well as was expected. Considering the results of the other methods (as discussed above), this shows that these methods are not 'bound to fail by construction', but rather that they can fail in certain circumstances. Yet, what we would like to point out is that the referee's explanation of TDA+ QDM is not the best example of how the methods could be 'bound to fail by construction'. Although the application of TDA results in a bias on the number of dry days, this is a bias of only 1% on the total number of dry days in the time series considered (38 and 41.15 dry days on the observed 3470 dry days).

**Comment:** The third point is that as you can read above it is relatively easy to explain most of your results. Yet you do not even try to do that. Just presenting the results without explaining them will not help others to better combine bias adjustment methods in the future.

**Response:** We admit that, although we tried to give a mechanistic overview, this was not extensive enough. Hopefully our additions, guided mostly by the specific comments, now clarify more and give insight in the interplay between the different methods. In addition to the expansions in the Results Section, we have extensively rewritten the Discussion and Conclusions Section, so they better reflect the points raised in the Results and provide some guidelines based on these results for a better application of occurrence-bias-adjusting methods.

**Specific Comments**

**Comment:** l 157: I think you should mention that your Thresholding method is not able to account for a climate change signal in the dry-day frequency.

**Response:** We agree with this remark. A sentence has been added at the end of the original l.150

*To adjust the number of wet and dry days, the frequency of dry days in the observations and in the simulations is calculated. The difference between the two frequencies, $\Delta N$ is the number of days of the simulation time series that has to be adapted. This adaption is done by sorting the wet days in increasing order of precipitation amount and thus only changing the $\Delta N$ lowest days of the simulation time series by setting them to 0. The use of a historical $\Delta N$ entails an important drawback, as it is unable to account for a climate change signal in the dry-day frequency.*

**Comment:** l 165-167: Vrac et al. (2016) set Pmin = 8:64 10$^{-4}$ mm/day. That is much lower than your P$_{min}$ value and I am sure that this influences your results.

**Response:** Our use of 0.1 mm as a dry-day threshold and the accordingly higher value of Pmin were based on two aspects. 1) to have some consistency throughout the article and with the observational time series(i.e. the resolution of the observation series is 0.1 mm), 0.1 mm was used as the minimum dry-day threshold in all methods. 2) Vrac et al. (2016) also mention the use of different possible dry-day thresholds in the last paragraph of Section 2.1. They tested thresholds of 0.5 mm and 1 mm, which did not change the conclusions of the paper. As such, we assumed that using the lower threshold of 0.1 mm would not influence the results as well, as applying thresholding before using SSR would automatically imply that the Pmin value should be higher than the threshold value. However, based on your comment we investigated this and it indeed has an influence on the occurrence results. We expanded the text in Sections 3.1.2, 3.2.2 and the Discussion. Yet, the impact of this on the discharge is limited, and the results could still be generalized, with the reminder that parameter settings too can have an influence.

(Section 3.1.2) *This implies that the final step of SSR, the reintroduction of dry days, reintroduces too few dry days. As the final step of SSR is fully based on the resulting number of days below $P_{min}$ the number of dry days is supposed to be caused by the combination of SSR and QDM. Whereas QDM normally transforms wet days into dry days based on the ratio of the time series' quantiles, as mentioned earlier, the first step of the transformation by SSR reduces the range of possible values that the lowest quantiles will have. Consequently, the difference between the quantiles will not be large enough to induce a change from wet to dry days by QDM, resulting in less days with a precipitation amount < $P_{min}$.*

(Section 3.2.2) *This changes the structure of dry day sequences and thus implies an extra change in the temporal structure on top of the effect of the Schaake Shuffle. Yet, as mentioned in Section 4.1.2, this extra change is influenced by the interplay between SSR and QDM.*

(Discussion) *Yet, despite the robustness of thresholding, some aspects should be considered. First, both SSR and TDA were influenced by the parameter settings. For the former, the dry day threshold had a larger influence than was expected based on Vrac et al. (2016). For the latter, the triangular distribution threshold b influenced which wet days could be transformed into dry days. On the one hand, this implies that better results could have been attained by other parameter settings. On the other hand, this could still interfere with the intensity-bias-adjusting method, and*

*especially with multivariate bias-adjusting methods. Nonetheless, even with better results, SSR or TDA can hardly have a higher added value than the direct or thresholding approach. For those two methods, for three out of for precipitation occurrence indices, the RB_MB values are closer to zero or zero (Fig. 4).*

**Comment:** l 173: Why is SSR not also applied month by month like Thresholding and TDA? Say so if that would not make a difference.

**Response:** SSR is not applied on a monthly basis as we wanted to keep the original setup as much as possible. Nowhere in Vrac et al. (2016) the use of SSR on a month-by-month basis is mentioned (consider also the use of a single Pmin value instead of monthly Pmin values). Neither does it make a difference if the threshold is calculated on a monthly value. Nonetheless, to clarify the difference between the methods, extra sentences have been added at the end of the original l 171.

*By using this method, the subsequent intensity-bias adjustment will be applied on all days, as if they were all wet days. There are thus no longer dry days, or singularities, in the time series, hence the method's name. In contrast with the thresholding and TDA methods, this method is not applied on a month-by-month basis. This is done to keep the original setup of Vrac et al. (2016) while it does not have an impact on the results.*

**Comment:** l 175: What is the motivation behind TDA? You only explain how the method works but not why (with which intentions) it was developed. What can it do better than other methods?

**Response:** TDA was developed during the PhD of M. T. Pham (Pham, 2016), at a time when there were no other methods available that could both add and remove dry days in a stochastic manner. It is based on the idea that not only the n driest days should be removed, in combination with a distribution that allows to remove days with a high precipitation amount, but at low probabilities. This has been implemented with a triangular distribution, but could be implemented using other distributions as well. This method had not been officially published anywhere else, and at the start of this research, we supposed it would be interesting to compare two recent stochastic methods (TDA and SSR). To clarify the intentions for developing this method, we altered the original l 175.

*The Triangular Distribution Adjustment method (TDA) is a rather complex method that was developed in an attempt to have a method that could easily both add and remove dry days while being stochastic (Pham, 2016). The main idea is that, by using a distribution, both stochasticity and a threshold to preserve precipitation can be introduced. Although the threshold is quite high, the probability that days becomes dry decreases rapidly based on the precipitation amount. In the first step, the number of days to be removed from or added to the future time series is calculated using the ratio of the historical observation and simulation dry day frequency respectively $f^{ho}$ and $f^{hs}$.*

**Comment:** l 177f: What if that is not possible, for example if $f^{ho} = 0.9$; $f^{hs} = 0.3$; $f^{fs} = 0.5$?

**Response:** Thank you for this remark, this is an important point that was not discussed during the original development of the method. Though this is not a problem for our time series, we added some sentences to discuss this and give a possible solution at the end of the original l 179.

*This ratio is assumed to be the same for the future scenarios such that it can be used to calculate a corrected future dry day frequency, $f^{fa}$, which is in turn used to calculate the number of days $\Delta N$ to be removed or added. A shortcoming of this method is that $f^{fa}$ might become higher than 1 or lower than 0 if the bias between $f^{ho}$ and $f^{hs}$ were too large. In such cases, $f^{fa}$ should be bound to respectively 1 and 0, implying that the future month under consideration has either no or only wet days.*

**Comment:** l 194: Not clear to me: Do you loop through 1–3 one day at a time or how? I am confused because l 189f suggests something else.

**Response:** Indeed, for every dry day to be added, steps 1-3 are looped through. L 188-189 have been slightly altered to better reflect this.

*Dry days are added as follows, with steps 1-3 applied for all $\Delta N$:*

1. *Choose a day $t$ (with precipitation $x_t > 0.1$) randomly from the wet day time series.*

**Comment:** Fig. 2: This figure is currently not helping to understand the TDA method. Please add text to the caption that explains what you intend to illustrate here. What are the red and black arrows? Is this supposed to explain the dry-to-wet case or the wet-to-dry case? Is precipitation decreasing on the x axis in panel (a) and if yes then why? Why is xth < 1mm, should it not be bigger because b = 0.9? Is that supposed to be the CDF of the triangular distribution in panel (b) and if yes then why is the curve not smoother?

**Response:** The '1' in panel has been removed and the smoothness of the curve in panel (b) has been improved. Thank you for notifying these problems to use, those were artifacts from designing the figure. The red arrow displays how a value can be mapped to the triangular distribution and the black arrow displays this for the threshold value. The caption has been extended to clarify this.

*Overview of the distributions used in TDA: (a) CDF of the precipitation of the wet days in the future simulation, (b) CDF of the triangular distribution. The red line displays how a value $x^{fs}$ is transformed into a value $T(\xi)$ in the wet-to-dry case. The black arrow indicates the threshold value. Adapted from Pham (2016).*

**Comment:** l 218-220: There are indices i missing in the innermost brackets of both equations.

**Response:** Thank you for notifying us of this oversight, this has been updated.

**Comment:** l 221f: Which version is used for evaporation?

**Response:** The multiplicative version was used here. This was mentioned in another section of the article, but now here as well.

*In this paper, the additive version is used for temperature time series and the multiplicative for precipitation and evaporation time series. This choice is based on the work of Wang et al. (2014) who have shown that using the additive version for precipitation results in unrealistic precipitation values and introduced a multiplicative version. For evaporation, we follow the few available studies (e.g. Lenderink et al. (2007)) to use the same adjustment as for precipitation.*

**Comment:** l 223: You should also specify how many empirical quantiles you used in your QDM configuration.

**Response:** The number of quantiles was determined automatically by the Matlab function used. This has now been clarified in the text.

*These days were used to build an empirical CDF (as in Gudmundsson et al.(2012); Gutjahr and Heinemann (2013), among others), because of the ease of application. The number of quantiles implemented was determined automatically by the Matlab function ecdf.*

**Comment:** l 227-229: Which threshold was used for evaporation and why? What do you mean with "calculated"? Do you mean that the empirical CDFs in Eq. (6) where based on wet-day values only? Please be more specific.

**Response:** For evaporation, 0.1 mm was also used and as such, for both evaporation and precipitation, Eq. (6) was applied on 'wet'-day values only. 'Calculated' was a redundant word choice here, and has been removed. The rest of the text has been adapted to better reflect the application of QDM.

*It is also important to note that for precipitation, Eq. (6) was applied only on the days considered wet, i.e. with a precipitation higher than 0.1 mm. For consistency, a threshold of 0.1 mm was also used for evaporation. As no prior examples for evaporation adjustment were available, we assumed this consistency was the best option.*

**Comment:** 248-250: Does that mean that MBCn was only applied to days with precipitation > 0.1 mm and evaporation > 0.1 mm? Or do you mean that QDM in the last step of MBCn was only applied to days with precipitation > 0.1 mm and evaporation > 0.1 mm?

**Response:** This sentence is about the application of QDM in the last step. As the use of a threshold for evaporation has been better clarified earlier in the text (see the previous comment) and the specific application of QDM is also mentioned later in the text, this sentence has been removed.

**Comment:** l 255f: Cannon (2016) uses the additive version of QDM for that. Did you do the same?

**Response:** Yes. To clarify this, 'additive' has been added in the text.

**Comment:** l 270-285: This description is quite confusing. I think you do not need to mention the Schaake shuffle and you do not need Eqs. (11-16) because all you do in the last step is shuffle $X_{fa}$ along the time axis such that the rank time series become identical to those of $X_{fs[k]}$, where k is the number of iterations of steps one to three.

**Response:** In our opinion, the addition of the Schaake Shuffle Eqs. clarifies the shuffling along the time axis. However, it may indeed be that it is confusing, as only the general formulation of the Schaake Shuffle was used, and the link with the notation previously used was not clearly given. A sentence has been added after the equations to clarify this.

*In our case, $X^{fa}$ is sorted according to $X^{fs}[j+1]$ instead of respectively X and Y, resulting in a final adjusted dataset $X^{fa'}$.*

**Comment:** l 286f: Does that imply that MBCn was also applied using 91-day running windows etc.? You better be more explicit here.

**Response:** Yes, it does. The sentences have been updated to better reflect this.

*The version of QDM applied in the step prior to the Schaake Shuffle is the same as in Section 3.4. As such, the shuffling procedure is the only difference with the univariate bias-adjustment, implying that differences in performance can be related to it.*

**Comment:** l 295f: Cannon (2016) also suggests early stopping. Have you tested that? What did you do to detect and prevent overfitting?

**Response:** The mention of overfitting was also based on Cannon (2018). As the citation was already in the previous sentence, it was not mentioned here not to overburden the text. However, for clarity, it has now been added. As for your question regarding the overfitting itself, the computation was halted based on a tolerance for consecutive energy distances. This seemed to suffice in our situation, as the number of variables is low. Though this is mentioned earlier in the text, this has been moved to this paragraph and extended for clarification.

*Third, to avoid overfitting (and to reduce the computational cost), early stopping is also suggested by Cannon (2018) (e.g. Prechelt (1998)). As the number of variables is limited in this case, overfitting did not seem to be a problem. Yet, to reduce unnecessary computational costs, the similarity in consecutive energy distances was used as a measure to stop the computation. A tolerance of 0.0001 was used: if the difference between two consecutively calculated energy distances was lower, the computation was halted.*

**Comment:** l. 297: It would help the readers a lot (see below) if you inserted an extra section here that spells out how exactly you combine the dry-day frequency BA methods with QDM/MBCn, and how QDM/MBCn are applied when they are not combined with any dedicated dry-day frequency BA method. For example, in TDA+QDM, is TDA done before or after QDM? The same for Thresholding+QDM etc.

**Response:** We added a paragraph in Section 2.4 to clarify this.

*In all combinations, first the occurrence-bias-adjusting method is applied on the time series and subsequently the intensity-bias-adjusting method. Only in the case of SSR, the last step of the occurrence calculations is applied after the application of the intensity-bias-adjusting method. In the direct approach, the occurrence-bias-adjustment is skipped and either QDM or MBCn is applied without any pre- or post-processing.*

**Comment:** l 310: You also need to specify whether you account for seasonality or just pool data from all days of the year.

**Response:** The data from all days of the year were pooled. Although we admit that it would be interesting to compare seasonal effects, this would have made our analysis and results more burdensome, given the large number of indices used. To make this more clear, the original l 334 has been extended.

*After these multiple calculations were carried out, the value of each index was first calculated on the basis of every simulation, pooling all data from the time series.*

**Comment:** l 312: Why can that be misleading?

**Response:** The directly adjusted indices are also used to build the transfer function, and should thus be corrected by construction. This will allow to correct even unrealistic scenarios. Hence, to be more careful, indirectly adjusted indices should also be compared. The sentence has been altered to better reflect this, with the mention of cross-validation removed as it is not used in this study.

*They argue that only using directly adjusted indices can possibly be misleading, as those indices are used to build the transfer function in the historical setting and should be corrected by construction.*

**Comment:** l 321-327: Which variables are used to drive PDM? Precipitation only or precipitation, evaporation, and temperature? I assume all three variables because otherwise it would make no sense to apply MBCn.

**Response:** Thank you for this remark, this is indeed missing in the text. Precipitation and evaporation are used as inputs, and this has been added in the text. Though temperature is thus not an input, we also included it in MBCn as most multivariate bias adjustment methods are made to adjust it and precipitation. As such, it is more comparable with other studies using MBCn. The results for temperature were also used in another study, which is currently in preparation.

*Similar to Pham et al. (2018), we use the Probability Distributed Model (PDM, Moore (2007)), a lumped conceptual rainfall-runoff model to calculate the discharge for the Grote Nete watershed in Belgium. This model uses precipitation and evaporation time series as inputs to generate a discharge time series.*

**Comment:** l 327: What did you use as your true discharge? Did you use observed discharge or did you generate it running PDM based on the observed climate data?

**Response:** This is an oversight and should have been mentioned in the text. Observed precipitation and evaporation time series are used as drivers for the PDM, the resulting discharge time series is considered to be 'observed' discharge. The text has been altered and extended to represent this.

*To calculate the bias on the indices, observed, raw and adjusted RCM time series were used as forcing for this model. The discharge time series generated by using the observations is considered to be the 'observed' discharge, and biases are calculated in comparison with this time series.*

**Comment:** l 332: MBCn is also stochastic, or did you use fixed rotation matrices?

**Response:** While MBCn indeed has a stochastic step when using random matrices, as we did, we considered its stochasticity to be minor in comparison due to the convergence towards the same observed time series. This is in contrast with e.g. the R2D2 method (Vrac, 2018), which is truly a stochastic multivariate bias-adjusting method.

**Comment:** l 368: Which kind of extrapolation are you talking about here? There is no extrapolation necessary to make Eq. (6) work for all values of $x^{fs}$.

**Response:** This is based on the advice that is often given for other quantile mapping methods (see e.g. Li et al. (2010)) to extrapolate the observed time series when the largest quantile of the observed time series is smaller than the largest quantiles of the simulations. The text has been rewritten to make this clearer.

*However, in contrast to what is advised for many other quantile mapping methods (e.g. Li et al. (2010)), quantile delta mapping does not explicitly apply extrapolation on the observed time series if the highest simulated quantiles are larger than the highest observed percentile. *

**Comment:** l 380-396: It is impossible to understand these results if you do not spell out exactly how you combine the dry-day frequency BA methods with QDM and how QDM is applied when it is not combined with any dedicated dry-day frequency BA method.

**Response:** See the comment about l. 297

**Comment:** l 389f: Who assumes that? Do you have a reference or can you motivate that assumption? I would not have expected the stochasticity added by SSR and TDA to improve the time series.

**Response:** This is based on the discussions and implementations by Maraun (2013), Mao et al. (2015) and Vrac (2018), among others. This was mentioned in Section 3.3, though less strongly than it is at l 389f, and we did not repeat those references. The sentence has been rewritten and expanded to make it more conservative.

*Some authors comment that additional randomness, which is introduced by the TDA and SSR methods, might improve the bias-adjustment results (Maraun, 2013; Wong et al., 2014; Mao et al., 2015; Vrac, 2018), by creating an ensemble of bias-adjusted results and accounting for local unexplained effects. This could result in an 'ensemble mean' that is closer to the observations.*

**Comment:** l 403-407: Your discussion of those small differences raises the question of statistical significance. There are no uncertainty estimates provided anywhere in your results section. Either you change that (using bootstrapping or the uncertainty originating from the stochastic nature of SSR and TDA, for example) or you are more careful with such statements.

**Response:** We decided that adding extra statistical analyses to the results would add too much work and burden an already lengthy paper. However, you are right to comment that the statement is not careful enough, although we tried to be careful, given the full paragraph. The original l. 403-404 have been rewritten.

*There are some small differences between the occurrence-bias-adjusting methods, suggesting that the combination of the nonlinear PDM and the occurrence-bias-adjusting methods could increase the effect of the latter.*

**Comment:** l 421-423: I do not understand. Please rewrite.

**Response:** The sentence has been rewritten for clarity.

*In contrast with all other results, Fig. 7 illustrates that the combination of MBCn and occurrence-bias-adjusting methods performs worse than the raw climate simulation for many precipitation occurrence indices.*

**Comment:** l 423-428: This part is quite difficult to read. Please rewrite. You are just describing what can be seen in Fig. 7. This can be done more intelligibly.

**Response:** The sentences have been rewritten.

*The wet-to-dry transition probability and the autocorrelation are only slightly worse than the raw climate simulations, but the dry-to-dry transition probability has $RB_{MB}$ values above 1.5 for every combined method, as can be seen in Table 2. The only two exceptions to the generally poor performance are the wet-to-dry transition probability for SSR and the number of dry days for all methods. The former is contrasting with the results in other Sections, in which the SSR method often performed the worst. Despite the good performance for the wet-to-dry transition probability, SSR performed worst for the dry-to-dry transition probability, with values of 1.23 and 2.56 for respectively $RB_O$ and $RB_{MB}$.*

**Comment:** l 432: Do those ranges include results for SSR or not?

**Response:** No, we have made this more explicit in this sentence.

*Except for SSR, the influence of the occurrence-bias-adjusting methods on precipitation occurrence is negligible, with e.g. values for $P_{P10}$ for MBCn without specific adjustment, thresholding and TDA ranging from 1.03 to 1.04 for $RB_O$ and ranging from 1.07 to 1.09 for $RB_{MB}$.*

**Comment:** l 432f: Here you start discussing the influence of QDM on precipitation occurrence but then you just stop and do not follow up in the discussion section.

**Response:** The referee might have misinterpreted it, as this describes the different effects of QDM and MBCn on precipitation occurrence, rather than the influence of QDM. As described in other comments (l 495-501), the application of QDM has been clarified. To ensure the clarity of the text, the original l 432f has been updated.

*The mostly negligible differences between the occurrence-bias-adjusting methods imply that the performance is largely influenced by the intensity-bias-adjusting method. The only difference between the multivariate and the univariate intensity-bias-adjusting methods is the rank-based Schaake Shuffle, as previously mentioned, which clearly demonstrates that the shuffling is responsible for the difference in performance seen here.*

**Comment:** l 435: "might be responsible for the performance" and "plausible cause": Why so vague? The shuffling changes autocorrelation and transition probabilities but not the distribution. There is no doubt about that.

**Response:** Because of the combination of methods, we wanted to be more careful. But as you mention, the shuffling is the main source for the poor performance of the occurrence-related indices. The word choice has been changed to better reflect this.

*The only difference between the multivariate and the univariate intensity-bias-adjusting methods is the rank-based Schaake Shuffle, as previously mentioned, which clearly demonstrates that the shuffling is responsible for the performance seen here. As the shuffle changes the time series structure, it has a direct influence the poor performance of occurrence-related indices.*

**Comment:** l 436-438: Aha. You should mention this earlier, ideally in a dedicated section, see above.

**Response:** We added more information on the application of the methods. Yet, we still decided to keep these lines here, as they are also well suited for the Results.

**Comment:** Table 2: It is unfortunate that you do not just plot all index values in your figures. I suggest you just extend the axis limits and get rid of this table. Then all the information can be found in one place and in one format.

**Response:** The plot size is chosen to make the different figures easily comparable and to use the space as optimal as possible. Not all plots need extension, and the spread of RBMB and RBO values higher than 1 differs from plot to plot, which would thus create either inconsistency if only a few plots would be changed. Alternatively, it would become unreadable if all plots would be changed according to the largest plot. Hence, we would like to keep the plot size and the tables as it is.

**Comment:** l 445-448: Can you explain that?

**Response:** The explanation is given a few lines below, which your next question also refers to.

**Comment:** l 450-452: Is that conjecture supported by Q5 being too high?

**Response:** Yes, although Q5 is 'not low enough' instead of 'too high', as it is still indicates a better performance than that of the raw climate simulation. The paragraph has been updated to be more clear and have a better flow.

*In contrast with the small effect of occurrence-bias-adjusting methods, MBCn itself seems to have a larger influence. The performance of MBCn combined with the occurrence-bias-adjusting methods for the lowest quantile shown here is worse than for the same quantile in Fig. 5. As mentioned in Section 2.2.5, the reordering by the Schaake Shuffle is the only difference with QDM. Therefore, it can be assumed that changing the time series structure by the rank-based reordering breaks sequences of low precipitation amounts, which results in less or different low flow sequences. This effect, which was clearly visible for the precipitation occurrence indices, makes it harder to adjust the already relatively small bias of the discharge. In contrast to the 5th quantile, the resulting performance for the combinations under consideration is better for the highest quantiles, for both $RB_O$ and $RB_{MB}$. This indicates that the climate model bias was large and most of this bias could be and has been removed by the bias-adjustment.*

**Comment:** l 452 You keep writing "5th quantile" where I think you should write "5th percentile".

**Response:** You are correct, this is a clear oversight. This has been changed in the text (not shown here as these are many small changes).

**Comment:** l 457f: Again, why would you assume that?

**Response:** See earlier comments and responses. In accordance with those responses, the sentence has been updated to be more conservative.

*Thus for this case, the assumption that the methods with randomized steps could improve the results by creating an improved 'ensemble mean' does not hold, as was the case for the univariate discharge percentiles in Section 3.1.3.*

**Comment:** Table 3: What is the point of this table? All of this can be seen in Fig. 8, so I suggest to remove it.

**Response:** We wanted to provide this table to give a more quantified result on the spread of the indices. However, as you say, this can also be seen in Fig. 8, albeit in less detail. Therefore, the table has not been removed, but has been moved to the appendix.

**Comment:** l 461-465: I think you are right but this can be spelled out more: SSR and TDA change precipitation ranks randomly and the partly randomized ranks are then shuffled by MBCn. Then PDM calculates discharge, the result of hydrological processes over many days. One could say that discharge integrates precipitation (an other fluxes) over space and time. Therefore, changing the temporal order of events has an effect on discharge. The impact on the high discharge quantiles is particularly strong for TDA (Fig. 8 vs. Fig. 5) because by setting b = 0.9, you have configured TDA such that it may turn pretty heavy precipitation days into dry days.

**Response:** Thank you for the suggestions. Some aspects of what you suggest have been added to the text. Your comment on TDA has also been added, as this is indeed a clear effect.

*It is unexpected that these are influenced by the occurrence-bias-adjusting methods, as these methods are supposed to only change the low precipitation amount values. However, the combination of (potentially random) changes in low precipitation amount values with the rank-based Schaake Shuffle and the non-linear integration of precipitation in the rainfall-runoff model might cause this effect. As the PDM integrates precipitation and evaporation fluxes over time, changing the rank structure evidently also influences the higher discharge percentiles. This effect of the shuffle becomes more clear in comparison with Fig. 5, which includes the same steps, except the Schaake Shuffle based on the preceding rotations. The effect on the high discharge quantiles can be exacerbated depending on the choice of occurrence-bias-adjusting method. In case of TDA, with b = 0.9, the higher percentiles are influenced more than by the other methods, as days with relatively high precipitation amounts can also become dry.*

**Comment:** l 466: Why do you cite Switanek et al. (2017) here? What in that paper is related to your statement?

**Response:** Thank you for remarking this mistake.

**Comment:** l 467: I think you should show those figures.

**Response:** The figure is now shown in the article. Combining this and your previous comment, some sentences have also been updated.

*As mentioned above, the highest percentiles are the most sensitive to these structural changes. This sensitivity is clearly visible when comparing the CDFs of the combinations under consideration (Fig. 9).*

**Comment:** l 469-472: You are talking about quite an extreme quantile here. Such a quantile can only be estimated with limited precision. You really have to quantify the statistical uncertainty of your results and measure the significance of differences if you want to make statements like these.

**Response:** We agree that more care should be taken with our statements. A few sentences have been added to better frame this. As mentioned earlier, we do not think that quantifying this statistically will make the reader more informed.

*For example, the 99.5th percentile performs much better for thresholding than for the other methods, with values for $RB_O$ and $RB_{MB}$ close to zero. However, this is a very high percentile, which is hard to estimate correctly. Nonetheless, the methods with added randomization and more repeats do not perform better, and thus do not seem to bring any added value.*

**Comment:** l 473-476: I am unable to see in Table A2 what you write here.

**Response:** This is a description of the differences in biases for the highest quantiles in Table A2 (now Table B2). However, the description was not clear enough and has been changed.

*Table B2 shows what the poor performance of the highest percentiles means in practice: the bias in the climate simulation is an overestimation. All methods correct this overestimation, but only thresholding does not adjust too much. The results of thresholding indicate that its temporal structure is the closest to the observations and that the temporal structure of the other methods does not allow the precipitation to build up the antecedent conditions for extreme floods.*

**Comment:** l 482f: Also about this I would have liked to read a deeper discussion.

**Response:** Although it is unclear what exactly you would have wanted to be discussed in more depth, we agree that there were some aspects left unmentioned. We extended this paragraph with a deeper discussion, linking some mechanical aspects of the methods to the discharge values. To add to this discussion, a small extension about how ties were accounted for has been added to the original line 286. We also admit that discharge was not mentioned in the Discussion Section. Based on this discrepancy with the actual importance of discharge for our research, we have decided to extensively reorder and rewrite the Discussion, with the result that discharge is now central to the arguments therein.

(Section 2.2.5 – l 286) *To account for ties in this procedure, a small random value is added before calculating the ranks of $X^{fs}_{[j+1]}$.*

(Section 3.2.3) *This difference in extreme flood causes can be additionally clarified by the remaining biases for $Q_{99.5}$ and $Q_{T20}$ in Table B1 and Table B2. For QDM, these are positively biased: the combination of antecedent conditions and the highest rainfall peaks causes discharge values that are too high, even though the highest rainfall extremes are negatively biased. As most of the remaining precipitation and precipitation occurrence biases are small, these biased discharge values are likely caused by remaining biases on evaporation. In contrast, the highest discharge values are negatively biased for MBCn, although there is the aforementioned difference between thresholding and the other values. This implies that the antecedent conditions are needed to provide realistic flood conditions. However, one could expect that the antecedent conditions for MBCn and thresholding + MBCn would be rather similar, given the similarity in occurrence adjustment between QDM and thresholding. Yet, the extreme discharge values indicate they are not. This difference can be explained by the Schaake Shuffle. QDM will also adjust the exact number of days to have unbiased dry days, but these 'dry' days will still have different values, although they are all below 0.1 mm. Consequently, the shuffle will impose a rank order on the dry days. As such, even though these days are all supposed to be equal, they are not for the Schaake Shuffle, which influences the temporal structure. On the other hand, thresholding will set all days that are supposed to be dry to 0 mm, which are indistinguishable for the Schaake Shuffle, thus retaining a slightly better temporal structure.*

**Comment:** l 489-490: That is not correct for all indices.

**Response:** Although the sentence was meant to imply that for some of the indices, the performance was worse, this should have been spelled out more, as you mention. This has been updated in the text.

*Although the results are good for discharge,* for *some of* the precipitation occurrence indices of the combinations of MBCn and the occurrence-bias-adjusting methods (Fig. 7), the performance is worse than the climate simulations.

**Comment:** l 495: You write that SSR+QDM and TDA+QDM performed "unexpectedly" worse. I think they did not, see my general comments.

**Response:** As indicated by the next comment and response, it was reasonable to expect SSR to perform well. We based our approach on the occurrence-bias-adjustment framework in Vrac et al. (2016) and other studies. As Vrac et al. (2016) indicated that it could be expanded to other quantile mapping methods, we assumed this to be an adequate framework for our study as well. Thus, rather than having inadequate methods, we have shown how these methods can fail in certain cases. The randomization does not necessarily imply a poor performance (as elaborated on in other responses), although it performed poorly in our situation. We suspect this is more dependent on the time series we wanted to adjust, than on the possible unjustified use of the methods. As such, although we agree with the general statement that we could have placed more emphasis on the mechanistic aspects (as we hope we now did), we disagree with the referee's comment.

**Comment:** l 495-501: What you do not mention here is that Vrac et al. (2016) used a much lower Pmin value (see above) and that they replaced zeros with random numbers from ]0; Pmin[. Then they apply CDFt, which by design adjusts the dry-day frequency, in contrast to QDM. The SSR method was designed to work well in combination with CDFt. You are not using SSR as intended by Vrac et al. (2016).

**Response:** Your first point has been answered in an earlier question. Your two other points, however, are incorrect. QDM is also able to adjust the dry-day frequency, as it can change wet days into dry days. For example, for a day with low, but >0.1 mm, precipitation, it can change the precipitation into a value <0.1 mm (which is considered dry in the index calculations), if ratio of the observed quantile vs. the simulated quantile is small enough. We agree that this was not clearly written, and the text has been expanded at the end of Section 3.4 and in Section 3.6 to better clarify this. To comment on the last point, in Section 3.2 of Vrac et al. (2016), the use of CDF-t in the study is discussed, and it is stated *"it is strongly expected that the results and the ranking of the methods for occurrences will be equivalent with other methods for intensity correction. Hence, only CDF-t is used here."* Nowhere in the article is explicitly given that SSR should only be used in combination with CDF-t and we are thus certain that it was safe to use in this study, given that QDM and MBCn are both also developed using the quantile mapping approach. However, we did not indicate well enough why we only used SSR and not any of the other methods surveyed by Vrac et al. (2016), and neither did we indicate that Vrac et al. (2016) suggested that methods other than CDF-t could also be used. These two aspects have also been updated in the article.

(Section 2.2.4) *It is important to note that although QDM is only applied on wet days, it can still transform low-precipitation wet days into days considered dry (e.g. with a precipitation amount < 0.1 mm) if the ratio in Eq. (6) is small enough.*

(Section 2.3) *All indices were calculated taking all days into account, instead of only calculating them on wet days. However, as described for the methods, days are considered dry or wet if the precipitation amount is respectively < 0.1 mm or > 0.1 mm. This distinction is needed for the calculation of the precipitation occurrence indices.*

(Discussion) *First, the occurrence-bias-adjusting methods that were proven to be outperformed by SSR by Vrac et al. (2016), the positive correction and threshold adaptation, were not used in this paper, as we assumed they would still be outperformed. Second, another univariate method, CDF-t (Michelangeli et al., 2009) was used, but Vrac et al. (2016) expected other intensity-bias-adjusting methods to perform similarly good in combination with SSR. As such, the use of QDM enabled a better comparison with MBCn.*

**Comment:** l 506f: This has already been done, see for instance François et al. (2020,

doi:10.5194/esd-11-537-2020).

**Response:** We aware of this article. As this manuscript was submitted shortly before the publication of François et al. (2020) to ESD Discussions, it was not included yet. However, you are right that it should be referenced. We discussed the results of François et al. (2020) in the Discussion.

*François et al. (2020) compared multiple multivariate bias-adjusting methods and concluded that they all perform relatively poor considering precipitation occurrence. As such, care should be taken when using multivariate bias-adjusting methods for climate change impact assessment, if the influence of occurrence matters.*

**Comment:** l 524f: That is easily explained by the drizzle effect which is of course perfectly adjusted by thresholding but not by SSR+QDM and TDA+QDM, see my general comments.

**Response:** We interpreted this comment as indicating a need for a better explanation in the Conclusions section. This has been implemented in the rewritten Discussion and Conclusions Sections. Besides, what we did not mention clearly enough is that the randomized methods do not necessarily perform worse. For most indices, they just do not add any value. Given their increased computational needs, this is also sufficient to discard them, at least for the circumstances given here. Added value is now mentioned more clearly in the rewritten Discussion and Conclusion Sections.

(Discussion) *These differences were most pronounced for the methods that apply randomization: they performed unexpectedly worse or at least did not bring any added value.*

(Conclusions) *As a third and last question, if occurrence adjustment seems necessary, the assumptions of the occurrence methods should be compared with the time series to be adjusted. When the occurrence bias is caused by an abundance of dry days, methods such as SSR may be necessary. In contrast, when the occurrence bias is caused by the drizzle effect, a simpler and more robust method such as thresholding seems the best option, in both the uni- and multivariate case, as more advanced methods did not add any value.*

**Comment:** l 525f: That needs an explanation!

**Response:** This statement is mostly based on the univariate occurrence indices, which are a clear example. This is based on the adjustment of wet days by QDM (see other responses) and the consequent effect on discharge. The mechanistic principle has been added in Section 3.1.2, and has also been addressed in the rewritten Discussion and Conclusions sections.

*This indicates that the climate models simulated the wet and dry days at the right positions in the time series, but with a positive bias (as displayed in Table B1. As the construction of QDM is based on this bias, it will automatically transform a part of the lowest wet days into dry days. For a time series with a realistic temporal structure, the bias in the occurrence indices is also decreased. Thresholding will perform similarly. In contrast, the additional randomness induced on wet days by TDA and SSR seems to change the temporally correctly simulated time series too much.*

**Comment:** l 526f: That statement is just wrong. MBCn has no influence on the performance of the dry-day frequency BA methods. All it does is shuffle the time series.

**Response:** This statement was unclearly written. What we tried to imply is the effect on the resulting combined performance and not on the performance of the occurrence-bias-adjusting method as such. This has been addressed in the rewritten Discussion section.

**Comment:** l 530-535: To also say something positive, I support those statements.

**Response:** See below

**Comment:** l 535-538: I support your call to always use the simplest possible method. Yet as outlined in the general comments, I am afraid your study is ill-designed to support it.

**Response:** Thank you for both these supportive comments. We hope that the explanations given in these comments and the adjustments to the paper clarify the study and its results.

**Comment:** Table A1: QDM alone does not only perfectly adjust Ndry but also PP00 and PP10. How is that possible?

**Response:** As clarified in earlier responses, QDM can transform wet days into dry days.

**Comment:** Tables A1 and A2: I suggest you use better column headings: QDM, SSR+QDM, TDA+QDM, Thresholding+QDM, and similarly for MBCn.

**Response:** Thank you for the suggestion, this would indeed be clearer and has been updated. The figure captions have been updated as well.

**References**

Cannon, A. J.: Multivariate quantile mapping bias correction: an N-dimensional probability density function transform for climate model simulations of multiple variables, Climate Dynamics, 50, 31–49, https://doi.org/10.1007/s00382-017-3580-6, 2018

Cannon, A. J., Sobie, S. R., and Murdock, T. Q.: Bias correction of GCM precipitation by quantile mapping: How well do methods preserve changes in quantiles and extremes?, Journal of Climate, 28, 6938–6959, https://doi.org/10.1175/JCLI-D-14-00754.1, 2015

Casanueva, A., Herrera, S., Iturbide, M., Lange, S., Jury, M., Dosio, A., Maraun, D. and Gutiérrez, J. M.: Testing bias adjustment methods for regional climate change applications under observational uncertainty and resolution mismatch, Atmospheric Science Letters, https://doi.org/ 10.1002/asl.978, 2020

Li, H., Sheffield, J., and Wood, E. F.: Bias correction of monthly precipitation and temperature fields from Intergovernmental Panel on Climate Change AR4 models using equidistant quantile matching, Journal of Geophysical Research: Atmospheres, 115, D10 101,https://doi.org/10.1029/2009JD012882, 2010.

Maraun, D.: Bias correction, quantile mapping, and downscaling: Revisiting the inflation issue, Journal of Climate, 26, 2137–2143,https://doi.org/10.1175/JCLI-D-12-00821.1, 2013.

Mao, G., Vogl, S., Laux, P., Wagner, S., and Kunstmann, H.: Stochastic bias correction of dynamically downscaled precipitation fields for Germany through Copula-based integration of gridded observation data, Hydrology and Earth System Sciences, 19, 1787–1806,https://doi.org/10.5194/hess-19-1787-2015, 2015.

Pham, M. T.: Copula-based stochastic modelling of evapotranspiration time series conditioned on rainfall as design tool in water resources management, PhD thesis, Faculty of Biosciences Engineering, Ghent University, 2016.

Vrac, M.: Multivariate bias adjustment of high-dimensional climate simulations: the Rank Resampling for Distributions and Dependences (R2D2) bias correction, Hydrology and Earth System Sciences, 22, 3175, https://doi.org/10.5194/hess-22-3175-2018, 2018.

---

## Author Comment (AC3) · 7 Oct 2020

**Response to Referee Comment 2 on 'Comparison of occurrence-bias-adjusting methods for hydrological impact modelling' by J. Van de Velde et al.**

**Anonymous referee**

We would like to thank the referee for the time spent reviewing our manuscript. Below, we give an overview of the comments, our responses and textual changes. For a detailed in-manuscript overview, we refer to the accompanying file.

**General Comments**

**Comment:** This paper compares different bias adjustment (BA) schemes for climate data as used in hydrological impact studies. With respect to precipitation, the originality resides in the fact that intensity and occurrence BA methods are used. Also, univariate and multivariate schemes are compared. The analysis is based on a high-quality climate data set, yet only a 20 year window is used to make the comparisons. For the comparison, precipitation amount, precipitation occurrence and discharge are evaluated. The results suggest that for the randomness inducing BA methods, performs worse than traditional methods. The results also show that the multi-variate methods lack robustness with respect to the modelling of precipitation occurrence and change the intrinsic structure of the time series in an unexplained way. The results suggest also that the performance of the current analysis is case specific and lacks generalization (the analysis should be repeated for climate series of other climate regions, and in the case of the multivariate method probably also with other climate variables). The paper is very detailed and uses state-of-the-art methods for implementing BA and for comparing novel BA approaches. The paper is also novel and original as it compares different BA methods for a specific climate region using a high-quality data set. Yet the paper lacks focus and is difficult to read. For instance, the BA is on climate data but the authors report also on the impact of climate data on modelled discharge. While this may be interesting to evaluate the impact of BA on final hydrological impact assessments, it does not add value to the comparison of the climate data in-se and only results in less focus of the manuscript. The final conclusions are also somehow disappointing: the addition of complexity in the BA did not result in better results. It is regretted that the reasons for these poor performances of more complex approaches are not explained. The reader is left with a feeling of : "So what? Why do the simple BA approaches outperform as compared to the more complex approaches". No real answers on this question has been offered by the authors.

**Response:** We thank the referee for the suggestions and points raised. We would like to address them point by point. First, the referee discusses the lack of generalization. It is true that the paper is focused on one location, but this was an opportunity-based choice. This way, we could base our analysis on a high-quality observational dataset, instead of processed observations and/or simulations as is the case when using reanalysis datasets. Besides, the focus on one location allows us to zoom in and discuss some aspects that might get lost when averaging over a larger area. We have strengthened this perspective and added more generalization by rewriting the conclusion. It is now based on a few questions that could guide researchers or users in other areas in the assessment of occurrence-bias-adjusting methods. With this change, we wanted to also address a second point of the referee: the lack of focus. By rewriting the Discussion and Conclusions sections from a discharge perspective and generalizing this to raise a few questions with regards to the use of occurrence-bias-adjusting methods for climate change impact assessment, we hope to have a clearer and more cohesive ending of the paper, with a better link with the Introduction.

We decided not to remove the results and discussion on modelled discharge. Discharge is raised in many papers (as discussed in our introduction) as one of the most important reasons to adjust occurrence. Consequently, we decided that without discharge, the motivation for the article would be less clear and the result were harder to interpret. The focus on discharge is the reason as well

not to discuss other climate variables adjusted by the multivariate method. Precipitation is the most important driver for discharge, and the subject of all occurrence-bias-adjusting methods, thus we wanted to only focus on this variable. However, in an article in preparation, we will discuss the effect of multivariate bias-adjusting methods on other variables in a similar framework.

A last, and important point, is the 'so what'-feeling. This issue was also raised by the other referee, and based on his comments, we have largely expanded our discussion on the (interaction between the) methods. Whereas the text in the Section 3, on the methods themselves, has been expanded slightly, most additions have been made to the Results section, were we have tried to discuss in depth why certain issues have arisen and what their effects are. For more details, we refer to Referee Comment 1 and the adapted paper. This point also leads to the aforementioned changes to the Discussion and Conclusions: the reasons for certain methods performing well or poor, have been integrated into a series of questions that can be used as a decision tree for occurrence-bias-adjustment.

**Specific Comments**

**Comment:** Title: The title could be reformulated to focus on the climate data

**Response:** Given the reasoning in response to the general comments, we have not removed discharge from the paper. Therefore, we prefer to keep the title unchanged.

**Comment:** It is suggested to create a clear "material and methods" section.

**Response:** We have followed this suggestion. The new 'Data and methods' section bundles the former 'Data' and 'Bias-adjusting methods' sections.

**Comment:** Line 80 – 103. This section should be moved to a "material and methods" section.

**Response:** We have merged l. 92-103 with the relevant parts of the 'Data and methods' section, but the l. 80-91 felt relevant enough to be in the introduction, as it also included our motivation for using certain methods. This seemed to general for inclusion in the 'Data and methods' section.

**Comment:** Line 105 – 109 is not needed.

**Response:** These lines have been removed.

**Comment:** Line 310 -314. This section can be removed.

**Response:** As this is related to the discharge, these lines were not removed.

**Comment:** Table 1. Index 2 & 3 could be removed.

**Response:** As this is related to the discharge, these lines were not removed.

**Comment:** Line 321-328 could be removed.

**Response:** As this is related to the discharge, these lines were not removed.

**Comment:** Line 338-357. This section could be moved to a "material and methods section".

**Response:** This has been adjusted. These lines are now included in Section 2.3 'Evaluation', which is part of Section 2: Data and Methods.

**Comment:** Fig.5, Fig. 8. The quality of the symbols should be revised.

**Response:** This was a technical problem. The quality of these symbols and those of the other figures as well has been updated.